# Communication-Efficient Federated Learning with Adaptive Number of Participants

## Abstract

While communication efficiency is a central challenge in Federated Learning (FL), standard protocols typically rely on a fixed, heuristically chosen number of participating clients per round. This rigid approach often leads to redundant communication in easy optimization stages or insufficient aggregation in heterogeneous regimes. In this work, we propose Intelligent Selection of Participants (ISP), an adaptive algorithm that dynamically optimizes the number of active clients to maximize communication efficiency without compromising convergence. Theoretically, we derive a convergence bound for the non-convex setting, revealing that the required number of participants scales with the gradient heterogeneity, rather than the total number of devices in the network. Guided by this insight, ISP speculatively adjusts the participation budget based on real-time training dynamics. ISP achieves consistent communication savings of up to 30% while matching the final accuracy of full-budget baselines. Furthermore, detailed ablation studies highlight the robustness of our adaptive criterion, establishing the dynamic selection of client count as a critical, distinct optimization task in federated systems.

## 1 Introduction

The rapid scaling of machine learning models and datasets has created unprecedented computational demands and data access challenges (Alzubaidi et al., 2021). Distributed setting addresses these constraints by enabling collaborative model training. Among distributed strategies, federated learning (FL) has emerged as a key paradigm (Kairouz et al., 2021; Konečný et al., 2016; Li et al., 2020a). It enables multiple decentralized data owners to collaboratively train models without directly sharing their raw data. Despite its numerous promises across a wide range of tasks (Smith et al., 2017; McMahan et al., 2017; Verbraeken et al., 2020), this paradigm presents unique challenges stemming from real-world deployment scenarios (Kairouz et al., 2021; Wen et al., 2023).

Among these challenges, communication efficiency emerges as a central bottleneck. Model updates must be transmitted over constrained and often unreliable uplink bandwidth. Communication complexity in FL scales directly with the number of participating clients, creating a fundamental trade-off. While involving more clients accelerates convergence through enhanced gradient diversity, it dramatically increases per-round communication overhead. This may overwhelm limited resources or trigger participant dropout due to resource limitations. Conversely, engaging too few clients slows learning dynamics and compromises model generalization.

While client selection strategies have been extensively studied, the optimal number of clients per training round remains largely underexplored despite its critical impact on scalability. Existing FL methods such as FedAvg (McMahan et al., 2017) and FedProx (Li et al., 2020b) assume a fixed number of participating clients per round, overlooking another optimization oportunity. Our core insight is that the participant count can and should be regarded as a dynamic variable to balance communication overhead against convergence speed. We propose the *Intelligent Selection of Participants* (ISP) approach, which dynamically determines the number of clients needed to achieve meaningful progress in each training round.

## 2 RELATED WORK

**Classic federated algorithms.** The scaling of model architectures and training data has stimulated adaptation of optimization algorithms to distributed environments. Built on the foundations of distributed synchronous SGD (Chen et al., 2017), the seminal Federated Averaging (FEDAVG) algorithm (McMahan et al., 2017) alternates local client training with global model averaging to trade-off computation and communication. Subsequent work has addressed data and system heterogeneity through various mechanisms. FEDPROX (Li et al., 2020b) added a proximal term constraining local updates, SCAFFOLD (Karimireddy et al., 2020) controled variates to correct client–server drift, FEDNOVA's normalized update magnitudes across variable local steps (Wang et al., 2020). Thus, they ensured fair aggregation regardless of computational differences between clients. Furthermore, (Reddi et al., 2020) applied adaptive Adam-like optimization to enhance efficiency. Comprehensive surveys like (Li et al., 2020a) covered a wide range of algorithms, showing the intensive development of this direction. While these advances have narrowed the performance gap to centralized training, they still rely on synchronized parameter exchanges between all clients and the server.

**Clients sampling.** Communication efficiency in federated learning can be significantly improved by carefully selecting participating clients. Early approaches employed static policies based on statistical contribution (Cho et al., 2020; Lai et al., 2021) or system utility (Nishio & Yonetani, 2019; Xu & Wang, 2020; Ribero et al., 2022). Instead of uniform sampling, client selection based on local loss values, which required prior communication, was proposed (Goetz et al., 2019). To mitigate this, POWER-OF-CHOICE (Cho et al., 2020) sampled a subset of clients for the loss evaluation. FED-COR (Tang et al., 2022) further reduced communication complexity by approximating loss changes using a Gaussian process. Statistical techniques like Importance Sampling (IS) Rizk et al. (2022) ranked clients by the variance, while DELTA (Wang et al., 2023) advocated for selecting clients with diverse gradients. FED-CBS (Zhang et al., 2023) sorted clients based on the label distribution uniformity. Hybrid strategies like OORT (Lai et al., 2021) and PYRAMIDFL (Li et al., 2022) integrated statistical and system metrics, often assuming a fixed number of clients per round. Nevertheless, all these works rely on a fixed number of selected clients.

**Number of clients choice.** There is, indeed, a small number of papers that explored dynamic control over the number of clients participating in a communication round. The authors of (de Souza et al., 2024) proposed an exponential decay schedule to gradually reduce the number of clients. This approach can negatively affect convergence in cases of statistical heterogeneity, since the model becomes increasingly sensitive to local shifts in client updates as it approaches the optimum. Other studies, such as (Chen et al., 2022), examined how the number of clients influences fine-tuning of large transformer models in federated settings. However, they do not assess dynamic schemes for selecting the number of clients and is aimed exclusively at a specific scenario of transformer fine-tuning. Most relevant is Chen et al. (2021), who directly addressed round-wise client count selection through ADAFL. However, their approach relies on a simple linear scheduling heuristic, which, unlike (de Souza et al., 2024), exhibited a positive trend.

Despite these efforts, existing approaches lack principled methods for determining optimal client participation based on real-time training dynamics, data heterogeneity, and resource constraints. We address this limitation by proposing the first dynamic approach that intelligently adapts client count based on training progress indicators rather than relying on predefined schedules.

**Main contributions.** We summarize the key contributions of this work as follows:

- **Adaptive participant-count selection.** We formalize the round-wise choice of the number of participants $m$ as a constrained problem. To solve it, we propose ISP (Intelligent Selection of Participants) that select the smallest $m$ that achieves expected loss decrease, explicitly reducing communication complexity.
- **Practicality and integration.** The procedure is compatible with popular federated algorithms such as FEDAVG and SCAFFOLD. It requires no changes to client optimizers and integrates with standard sampling and aggregation, amortizing the auxiliary synchronization overhead across subsequent rounds.
- **Empirical verification.** We conduct extensive and reproducible experiments on diverse benchmarks, including synthetic CIFAR-10 (Krizhevsky et al., 2009) and Tiny-

IMAGENET (Le & Yang, 2015) datasets, as well as a real-world electrocardiogram (ECG) classification task. Our code is publicly available[1]. ISP achieves up to a 30% reduction in communication rounds over already efficient client-sampling and gradient-compression techniques while maintaining or improving accuracy. Our ablations cover the number of sampled subsets used by the estimator, the interval between client-count updates, and the momentum/smoothing parameters (see Appendix E).

## 3 METHOD

### 3.1 PROBLEM SETUP

We consider the global training objective as the solution to

$$\min_{\boldsymbol{x} \in \mathbb{R}^d} \; f(\boldsymbol{x}) := \sum_{i=1}^{M} w_i \, f_i(\boldsymbol{x}), \tag{1}$$

where $f_i(\boldsymbol{x}) = \mathbb{E}_{\boldsymbol{z}_i \sim \mathcal{D}_i}[f_i(\boldsymbol{x}, \boldsymbol{z}_i)]$ is the expected loss of client $i$ under distribution $\mathcal{D}_i$, $\boldsymbol{x}$ are the model parameters, and $w_i$ is the client weight. We write equation 1 in weighted form to capture non-uniform sampling: e.g., in FEDAVG, $w_i = n_i/n$, with $n_i$ local samples and $n = \sum_i n_i$ total, in OORT $w_i = p_i$ reflect the overall (statistical and system) utility score. To systematically analyze such sampling behavior in FL, we introduce the concept of a client sampling strategy, formalized below. This will allow us to unify a broad class of methods under a common mathematical structure.

**Definition 3.1** (Client Sampling Strategy). A client sampling strategy $\mathcal{S}_\tau$ at round $\tau$ is a (possibly randomized) procedure that returns a subset

$$C_{\tau+1} \sim \mathcal{S}_\tau, \quad C_{\tau+1} = \{i_1, \ldots, i_{m_{\tau+1}}\} \subseteq \{1, \ldots, M\},$$

where $m_{\tau+1} = |C_{\tau+1}|$ is the number of selected clients.

This formalism provides a basis for describing and comparing different sampling mechanisms, whether uniform, deterministic, or adaptive.

*Example* 1 (Uniform Sampling). In the uniform sampling strategy, we randomly select clients without replacement:

$$C_{\tau+1} \sim \mathrm{Unif}\left(\{\text{all subsets of size } m\}\right).$$

This ensures unbiased gradient estimation, but does not adapt to variations in client data, compute resources, or connectivity.

*Example* 2 (Active FL). In Active FL (Goetz et al., 2019) sampling procedure, each client $i_{k_\tau}$ at round $\tau$ is sampled proportionally to its valuation $p_k$:

$$C_{\tau+1} = \{i_{k_\tau} \mid i_{k_\tau} = 1\}, \quad i_{k_\tau} \sim Be(p_k),$$

where $p_k$ proportional to the reduction of the client's loss.

Valuation $p_k$ represents the adjacency from client to server. It can also be set based on other utilities, such as the importance of the dataset.

Since the purpose of adapting the number of participants is communication savings, we specify this concept based on the fundamental principles of the FL procedure (Konečný et al., 2016; McMahan et al., 2017).

**Definition 3.2** (Communication costs). We define communication costs as the uplink transfer of model state from clients to the server.

$$Comm(\tau) = \sum_{i \in C_{\tau+1}} \mathrm{size}(\text{CLIENTUPDATE}(\boldsymbol{x}^\tau, f_i)),$$

where CLIENTUPDATE$(\boldsymbol{x}^\tau, f_i)$ performs local steps, returns the model, size$(\cdot)$ denotes its volume.

---

[1]https://anonymous.4open.science/r/ISPFL/

The size($\cdot$) emphasizes that communication costs are determined by the bottleneck of the client upload bandwidth. With this definition, we neglect the transfer of the scalar loss values (Problem 2) due to their negligible size. In the standard FL pipeline (Algorithm 1), the model state has fixed size, so we measure specific communication cost $Comm(\tau)/\text{size}(\boldsymbol{x}^\tau)$ by the number of CLIENTUPDATE calls. We report communication comparisons using this metric throughout the experiments.

**Definition 3.3** (Computation costs). We define computational costs as the client working time

$$Comp(\tau) = \sum_{i \in C_{\tau+1}} \left[ \text{time}(\text{CLIENTUPDATE}(\boldsymbol{x}^\tau, f_i)) + \text{time}(\text{CLIENTINFER}(\boldsymbol{x}^\tau, f_i)) \right],$$

where $\text{CLIENTINFER}(\boldsymbol{x}^\tau, f_i)$ performs an inference of $\boldsymbol{x}^\tau$ and returns the scalar loss value $f_i(\boldsymbol{x}^\tau)$, time($\cdot$) denotes its working time.

The computational cost takes into account the client processing overhead caused by model training and inference (Vogels et al., 2019; Wang et al., 2021). With this definition, we consider the CLIENTINFER burden, which is neglected in Definition 3.2. Computational costs were evaluated by measuring the mean duration of a training round throughout the FL training.

## 3.2 OUR METHODOLOGY

The Algorithm 1 formalizes the general federated learning setup, which serves as the basis for our later developments. A central server coordinates training across $M$ clients, each with its own dataset. At round $\tau$, it samples a subset $C_{\tau+1} \sim \mathcal{S}_\tau$ of size $m_{\tau+1}$. The selected clients update the global model $\boldsymbol{x}^\tau$ with a fixed number of local gradient descent steps on its own loss $f_i$. The server then aggregates these local states. After $T$ rounds, the final model $\boldsymbol{x}^T$ is obtained.

---

**Algorithm 1** Communication with Varying Client Set

**Require:** initial model $\boldsymbol{x}^0$, sampling strategies $\{\mathcal{S}_\tau\}$.
1: **for** $\tau = 0$ to $T - 1$ **do**
2:   $m_{\tau+1} \leftarrow ISP(\boldsymbol{x}^\tau, m_\tau, \tau)$     ▷ Algorithm 2
3:   Sample $C_{\tau+1} \sim \mathcal{S}_\tau$, $|C_{\tau+1}| = m_{\tau+1}$
4:   **for all** clients $i \in C_{\tau+1}$ **in parallel do**
5:     $\boldsymbol{x}_i^\tau \leftarrow \text{CLIENTUPDATE}(\boldsymbol{x}^\tau, f_i)$
6:   **end for**
7:   $\boldsymbol{x}^{\tau+1} \leftarrow Aggregate(\{\boldsymbol{x}_i^\tau\}_{i \in C_{\tau+1}})$
8: **end for**
9: **Output:** $\boldsymbol{x}^T$

---

In the classical FL paradigm, the number of participating clients $m_{\tau+1}$ remains constant across rounds. While this simplifies the analysis, it overlooks an important degree of freedom: varying $m_{\tau+1}$ can trade off communication cost against convergence speed. We can study this trade-off by introducing the *next-round loss changing*

$$\delta f_\tau(m_{\tau+1}) = f\left(\boldsymbol{x}^{\tau+1}(m_{\tau+1})\right) - f(\boldsymbol{x}^\tau), \tag{2}$$

where $\boldsymbol{x}^{\tau+1}(m_{\tau+1})$ is modeled according to random subsets $C_{\tau+1} \sim \mathcal{S}_\tau, C_\tau \sim \mathcal{S}_{\tau-1}$ of sizes $m_{\tau+1}, m_\tau$, respectively (see Line 7, Algorithm 1). The choice of $m_{\tau+1}$ governs the magnitude and direction of $\delta f_\tau$: larger values typically lead to more pronounced improvements in average loss, while smaller ones may even result in degradation due to insufficient update quality.

We address this trade-off by *minimizing* communication costs. Then, our goal is to choose the smallest $m_{\tau+1}$ that still ensures model improvement, i.e.,

$$\begin{aligned}
\min_{m_{\tau+1}} \quad & m_{\tau+1}, \\
\text{s.t.} \quad & \mathbb{E}_{C_{\tau+1}} \delta f_\tau(m_{\tau+1}) < 0.
\end{aligned} \tag{3}$$

Note that $\delta f_\tau(m_{\tau+1})$ also depends on random $C_{\tau+1}$, $\boldsymbol{x}^{\tau+1}(m_{\tau+1}) = \boldsymbol{x}^{\tau+1}(m_{\tau+1}, C_{\tau+1})$. Hence, the model improvement is implied as averaged. Solving equation 3 enables adaptive client selection, ensuring minimal resource use while maintaining next-round model improvement. By dynamically balancing $m_{\tau+1}$ with expected loss reduction, it reconciles efficiency and convergence in stochastic federated settings. We also consider an alternative optimization formulation (Appendix E.7). However, several challenges currently hinder equation 3 direct solution. We next examine these limitations and propose resolution strategies.

---

**Algorithm 2** ISP (Intelligent Selection of Participants)

---

**Require:** Iteration $\tau$, previous $m_\tau$, previous model $\boldsymbol{x}^\tau$.
    **Context:** Window $\Delta$, momentum $\beta$, resolution $w$, max clients $M$.
1: **if** $\tau \bmod \Delta \neq 0$ **then**                                  ▷ Run ISP each $\Delta$ rounds
2:     **return** $m_\tau$
3: **end if**
4: **for all** clients $i \in \{1, 2, \ldots, M\}$ **in parallel do**             ▷ Intermediate communication
5:     $\boldsymbol{x}_i^{\tau+1/2} \leftarrow ClientUpdate(\boldsymbol{x}^\tau, f_i)$
6: **end for**
7: $m \leftarrow 1$
8: **while** $m \leq M$ **do**
9:     $\delta f_{\tau+1/2}(m) \leftarrow \frac{1}{N} \sum_{n=1}^{N} f(\boldsymbol{x}^{\tau+1/2}(m, C_{\tau+1}^n)) - f(\boldsymbol{x}^\tau)$          ▷ Problem 2
10:     **if** $\delta f_{\tau+1/2}(m) < 0$ **then  Break**                         ▷ Early exit
11:     **end if**
12:     $m \leftarrow m + w$
13: **end while**
14: **return** $\lfloor \beta m + (1 - \beta) m_\tau \rfloor$

---

*Problem* 1 (Direct evaluation). The direct computation of $\delta f_\tau(m_{\tau+1})$ is infeasible, as the right-hand side of equation 2 depends on the future global model $\boldsymbol{x}^{\tau+1}$.

Since $\boldsymbol{x}^{\tau+1}$ is explicitly determined by a subset of clients $C_{\tau+1}$, the estimation of $\mathbb{E}_{C_{\tau+1}} \delta f_\tau(m_{\tau+1})$ requires updates from all workers. Thus, we need to approximate all possible realizations of the future global model $\boldsymbol{x}^{\tau+1}$. For this purpose, we propose a *full-client intermediate* communication round $\tau + 1/2$, i.e. $m_{\tau+1/2} = M$ and $C_{\tau+1/2} = \{1, 2, \ldots, M\}$. In this round, we will receive client updates $\{\boldsymbol{x}_i^{\tau+1/2}\}_{i=1}^M$, whose combination approximates $\boldsymbol{x}^{\tau+1}$ for an arbitrary $C_{\tau+1}$. The suggested intermediate communication leads to the following expectation

$$\delta f_{\tau+1/2}(m_{\tau+1}) := \mathbb{E}_{C_{\tau+1}} \delta f_{\tau+1/2}(m_{\tau+1}) = \mathbb{E}_{C_{\tau+1}} \left[ f(\boldsymbol{x}^{\tau+1/2}(m_{\tau+1}, C_{\tau+1})) - f(\boldsymbol{x}^\tau) \right]. \quad (4)$$

To maintain computational efficiency, clients can participate *partially* (i.e. $m_{\tau+1/2} < M$) or perform reduced local training epochs during this auxiliary synchronization phase (see Appendix E.1 for details). In any case, the current depiction of the federated learning pipeline assumes a solution equation 3 at each communication round (see Line 2, Algorithm 1), requiring regular full-client intermediate communications that degrade efficiency. To resolve it, we pose a dynamic number selection procedure as *strategy*: solve equation 3 once every $\Delta$ rounds and propagate the resulting $m_{\tau+1}$ to subsequent rounds. We also adopt a *$\beta$-momentum* term for updating $m_{\tau+1}$ (Line 14, Algorithm 2), blending the newly computed client count with past values to incorporate history and capture global trends. To reduce stochasticity in the equation 4 calculation, we apply exponential moving averaging to the $f(\boldsymbol{x}^\tau)$. For details on their impact, see Appendix E.2–E.4.

Intermediate communication and adaptive selection of their number as a strategy for the next $\Delta$ rounds allows us to formulate an optimization procedure of equation 3, which we call **Intelligent Selection of Participants** (ISP). This workflow is illustrated in Figure 1 and its pseudocode is represented in Algorithm 2. After intermediate communication, we loop over counts $m_{\tau+1}$, evaluate the objective in equation 4, and check its improvement. We now turn to the challenge of estimating this objective.

*Problem* 2 (Expectation estimation). The exact estimation of equation 4 requires enumerating all $\binom{M}{m_{\tau+1}}$ possible subsets. Thus, the exact solution of equation 3 is computationally prohibitive.

To estimate the expected value in equation 4, we use a *Monte Carlo* approach, consisting of repetitive sampling $C_{\tau+1}^n \sim \mathcal{S}_\tau$ and aggregating corresponding client updates

$$\boldsymbol{x}^{(\tau+1/2)^n} := \boldsymbol{x}^{\tau+1/2}(m_{\tau+1}, C_{\tau+1}^n) = Aggregate \left( \{x_i^{\tau+1/2}\}_{i \in C_{\tau+1}^n} \right).$$

We then obtain $f(\boldsymbol{x}^{(\tau+1/2)^n})$ loss by aggregating client local objectives $f_i$, computed via $f_i(\boldsymbol{x}^{(\tau+1/2)^n}) = \text{CLIENTINFER}(\boldsymbol{x}^{(\tau+1/2)^n}, f_i)$. This yields an estimate of equation 4 under suffi-

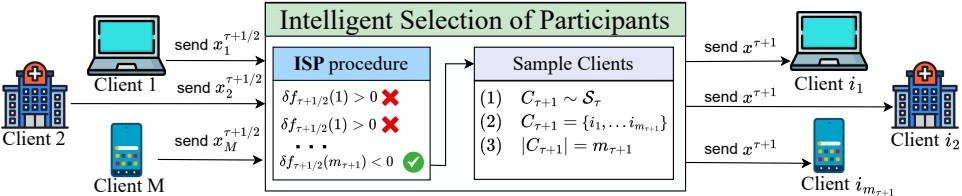

Figure 1: ISP procedure pipeline.

cient coverage

$$\delta f_{\tau+1/2}(m_{\tau+1}) \approx \frac{1}{N} \sum_{n=1}^{N} f(\boldsymbol{x}^{(\tau+1/2)_n}) - f(\boldsymbol{x}^{\tau}), \tag{5}$$

where $N$ defines the coverage depth, and is a method hyperparameter whose ablation study can be found in Appendix E.5. The Algorithm 3 in Appendix B encapsulates this proposition.

Despite the naivety of the approximation equation 5, this approach is not without meaning even in the case of significant client heterogeneity. This is largely due to the moderate variability of the sampling procedure $C_{\tau+1}^n \sim \mathcal{S}_\tau$, which is one of the central problems of the client selection strategies. For example, strategies POWER-OF-CHOICE (Cho et al., 2020) and FEDCOR (Tang et al., 2022) are deterministic, i.e. $N = 1$ gives an exact solution of equation 3. In FEDCBS (Zhang et al., 2023), the authors apply a sequential sampling strategy to reduce $N$, while in DELTA (Wang et al., 2023) it is directly proven that the proposed unbiased sampling scheme is optimal in terms of minimizing the variance of $\mathcal{S}_\tau$.

*Problem* 3 (Internal Computation). Optimization equation 3 requires enumerating $m_\tau = \{1, 2, \ldots, M\}$. Each enumeration through $m_{\tau+1}$ is associated with the estimating $\mathbb{E}_{C_{\tau+1}} f(\boldsymbol{x}^{(\tau+1/2)}(m_{\tau+1}))$, which requires calculating the loss of clients.

To reduce the search for $m_\tau$, we introduce a *resolution* factor $w$ that specifies the enumeration step and reflects the sensitivity of the method to the number of clients in a window of size $2w$ (see Line 12, Algorithm 2). For details on its impact, see Appendix E.6. Estimation $\mathbb{E}_{C_{\tau+1}} f(\boldsymbol{x}^{(\tau+1/2)}(m_{\tau+1}))$ requires repetitive sampling $C_{\tau+1}^n \sim \mathcal{S}_\tau$ and communication with clients to calculate $f(\boldsymbol{x}^{(\tau+1/2)_n})$. Although it does not pose a communication bottleneck (see Definition 3.2), multiple inferences can tax worker devices (see Definition 3.3). This limitation can be addressed by introducing a *surrogate* loss function $\hat{f}(\boldsymbol{x}^\tau)$, which is stored on the server and approximates client representations. This is a fairly strong but *optional* assumption that makes sense in the case of external publicly available datasets. We demonstrate its applicability in the Experiment 5.3 on real-world ECG data. We discuss computational impact of the surrogate loss in Appendix D.4. We also suggest how to relax this assumption in the Appendix E.8.

## 4 THEORETICAL ANALYSIS

We now provide a rigorous theoretical justification for Algorithm 2. We aim to show that our adaptive client selection strategy maintains the standard convergence guarantees of full-batch optimization while significantly reducing communication costs.

Our analysis relies on standard regularity conditions for non-convex optimization (Li et al., 2021; Richtárik et al., 2021). First, we assume the smoothness of the objective function.

**Assumption 4.1.** The function $f$ is $L$-smooth, i.e., $\|\nabla f(x) - \nabla f(y)\| \leqslant L\|x-y\|$ for any $x, y \in \mathbb{R}^d$.

Second, we address the non-convex nature of modern deep learning tasks.

**Assumption 4.2.** The function $f$ is **non-convex** if it has at least one (not necessarily unique) minimum, i.e., $f(x^*) = \inf_{x \in \mathbb{R}^d} f(x) > -\infty$.

Finally, we introduce an assumption quantifying the diversity of local gradients. To ensure convergence, it is necessary to bound the dissimilarity between individual client loss functions $f_i$. In the extreme case where local objectives are entirely disparate, full participation would be required at every step. However, in practical federated settings, local data distributions share common structures.

We assume data similarity in the following way:

**Assumption 4.3.** Clients possess $(\rho)$-heterogeneous loss functions for some $\rho \geqslant 0$, such that for all $x \in \mathbb{R}^d$, the following holds:

$$\|\nabla f_i(x) - \nabla f(x)\|^2 \leqslant \rho \|\nabla f(x)\|^2 \quad \forall i \in 1, \ldots, M.$$

This condition is also known as over-parameterization. Indeed, contemporary deep learning models, particularly Large Language Models (LLMs), operate in highly over-parameterized regimes (Vaswani et al., 2017; Chizat et al., 2019; Liu et al., 2022). Critically, in the federated setting, (Jian & Liu, 2025) demonstrated that over-parameterization induces a benign loss landscape where local gradients $\nabla f_i(x)$ of heterogeneous clients remain well-aligned with the global gradient $\nabla f(x)$ throughout training. This alignment is precisely what Assumption 4.3 captures: the bounded deviation $\|\nabla f_i(x) - \nabla f(x)\|^2 \leq \rho \|\nabla f(x)\|^2$ reflects that over-parameterized models naturally regularize gradient heterogeneity by operating in a regime where gradient noise is controlled relative to the signal. Consequently, the parameter $\rho$ quantifies data heterogeneity.

We first show that our procedure for selecting the number of devices via a Monte Carlo approximation and descent estimation indeed provides a descent in expectation. In particular, we consider equation 5. Let us formally define the expected descent at iteration $\tau$ given a dynamic $m_{\tau+1}$ as:

$$\mathbb{E}\delta f_\tau(m_{\tau+1}) = \mathbb{E}\left[\frac{1}{N}\sum_{n=1}^{N} f(x^{(\tau+1/2)_n}(m_{\tau+1}) - f(x^\tau(m_\tau))\right].$$

Without loss of generality, we fix $m_{\tau+1} = m_\tau = m$. Now we present a Theorem that demonstrates that our procedure ensures a strictly negative expected descent.

**Theorem 4.4.** *Under Assumptions 4.1, 4.2, and 4.3, Algorithm 2 with stepsize $\gamma \leq \frac{\alpha}{L}$ (where $\alpha < 2$) and adaptive batch size $m \geq \frac{\alpha\rho}{(2-\alpha)(M-1)+\alpha\rho}M$, guarantees a strictly negative expected descent in the Monte Carlo approximation of the objective function in Equation 5:*

$$\mathbb{E}\delta f_{\tau+1/2} = \mathbb{E}\left[\frac{1}{N}\sum_{n=1}^{N} f(x^{(\tau+1/2)_n}) - f(x^\tau)\right] < 0.$$

*Example* 3. *Substituting $\alpha = 1$ to set the step size $\gamma = \frac{1}{L}$ into the condition yields:*

$$m \geq \frac{\rho}{M-1+\rho}M \approx \frac{M\rho}{M+\rho}.$$

*This specific case illuminates the behavior of our selection criterion at the limits:*

- ***High Heterogeneity ($\rho \gg M$):** The condition requires $m \to M$, meaning full participation is necessary to control the variance when data is extremely non-IID.*
- ***Low Heterogeneity ($\rho \ll M$):** The requirement simplifies to $m \approx \rho$.*

This result implies that for a fixed step size, the required participation ratio $\frac{m}{M}$ decreases as the total network size $M$ grows, asymptotically scaling with the heterogeneity parameter $\rho$.

We now derive the global convergence rate, proving that this reduced communication strategy maintains the standard asymptotic efficiency of non-convex optimization.

**Theorem 4.5.** *Under Assumptions 4.1, 4.2, 4.3, after $T$ iterations of Algorithm 2 with $\gamma \leq \frac{\alpha}{L}$ (where $\alpha < 1$) and adaptive batch size $m \geq \frac{\alpha\rho}{(1-\alpha)(M-1)+\alpha\rho}M$, the averaged squared gradient norm is bounded as:*

$$\frac{1}{T}\sum_{t=0}^{T-1} \mathbb{E}\|\nabla f(x^\tau)\|^2 \leq \mathcal{O}\left(\frac{\Delta_T L}{T}\right).$$

**Discussion.** The derived $\mathcal{O}(1/T)$ convergence rate is optimal (Carmon et al., 2017). Moreover, from the communication perspective, our bound is tighter by a factor of $1 + \frac{M}{\rho}$, as our method requires participation from only $m$ out of $M$ devices. While the condition on $m$ incorporates a finite population correction, asymptotically (in the low heterogeneity regimes $\rho \ll M$), it simplifies to $m = \mathcal{O}(\rho)$, implying that the required active cohort size is determined primarily by data heterogeneity rather than the total network scale. This theoretical insight justifies our speculative strategy: it allows for substantial communication savings in regimes $\rho \ll M$ without degrading final quality.

We further extend this analysis to the stochastic setting (Theorem F.5) and the finite-sum mini-batch regime (Theorem F.6). All proofs and other details can be found in Appendix F.

## 5 EXPERIMENTS AND RESULTS

### 5.1 EXPERIMENT SETTINGS

To evaluate the impact of the proposed technique, we conducted extensive experiments on three datasets. The first two are public CIFAR-10 (Krizhevsky et al., 2009) and TINY-IMAGENET (Le & Yang, 2015), which represent the multiclass image classification task. The third proprietary dataset (due to privacy issues) consists of approximately 800,000 labeled 12-lead digital ECG segments 10 seconds long, sampled at 500Hz for patients older than 18 years. This dataset was collected in real-world clinical settings, and represents a time series multi-label classification. The independent test set contains approximately 400,000 ECG segments obtained under identical conditions. We utilize RESNET-18 (He et al., 2016) for CIFAR-10, visual transformer SWIN-T (Liu et al., 2021) for TINY-IMAGENET and an adapted RESNET1D-18 for multilabel ECG classification. The convergence of the model is controlled using the validation part of the local client data. The standard hyperparameters employed in our training across various scenarios are detailed in Appendix C, Table 6. We report communication costs as the number of CLIENTUPDATE calls (see Definition 3.2), with ISP intermediate communications included.

To demonstrate the capabilities of adaptive selection of the number of clients, all datasets were partitioned among them using the Dirichlet distribution parametrized by the concentration $\alpha$ (Hsu et al., 2019). This parameter controls the clients' statistical heterogeneity (strong when $\alpha \to 0$). Table 1 summarized the main hyperparameters. More technical details can be found in the Appendix C, D.

Table 1: Hyperparameter setup. CS – Client Selection (Table 2), GC – Gradient Compression (Section 5.4).

| Hyperparameters | Image domain | ECG domain |
|---|---|---|
| Number of Clients | 100 | 2000 |
| Dirichlet concentration | TINY-IMAGENET: $\alpha = 0.5$; CIFAR-10 (CS): $\alpha = 0.1$; CIFAR-10 (GC): $\alpha = 1000$ | $\alpha = 5.0$ |
| ISP-Solution ISP-Depth ISP-Resolution ISP-Momentum ISP-Surrogate | $\Delta = 20$ $N = 10$ $w = 1$ $\beta = 0.5$ No | $\Delta = 20$ $N = 20$ $w = 20$ $\beta = 0.5$ Yes |

### 5.2 DYNAMIC PARTICIPANTS WITH CLIENT SELECTION

**CIFAR-10 Experiments** The ISP methodology explicitly operates with the concept of a client sampling strategy and relies on it in expectation estimation (see Problem 2), making it natural to assess the impact of an adaptive number of clients on these strategies. We consider the baseline strategy of uniform selection of a random subset (McMahan et al., 2017), along with the following state-of-the-art techniques for client sampling mentioned in the related work:
• The POWER-OF-CHOICE (POW-D) (Cho et al., 2020) selects clients by local loss value within a random pool of size $D$. We fix $D = 40$ for this ranking;
• FEDCOR (Tang et al., 2022), obtains client losses, approximating them with a Gaussian process;
• FEDCBS (Zhang et al., 2023) enforce uniform class-based sampling, balancing exploration vs. exploitation with $\lambda = 10$;
• DELTA (Wang et al., 2023) in contrast select clients by maximizing external and internal diversities. We set the corresponding proportions $\alpha_1 = 0.8$ and $\alpha_2 = 0.2$.

The results are averaged over 5 runs and presented in Table 2. We set the baseline number of clients to $m = 20$, as motivated in Appendix D.3. We observe a significant increase in communication efficiency, up to 25% for FEDCOR and DELTA sampling strategies, while maintaining or even slightly improving the quality of the downstream task across all methods. This improvement is achieved even in spite of the full intermediate communication $m_{\tau+1/2} = M$. We further relax this condition in Appendix E.5. Only for the POW-D baseline, the ISP technique requires barely more communication but demonstrates a substantial increase in the target metrics that we attribute to the adversarial nature of client sampling in POW-D. The communication and accuracy dynamics for this comparison is illustrated in Figure 2a. Although we notice a minor time overhead (1–2%) per one communication round in certain configurations due to the selection logic in ISP, the technique significantly reduces training time across all baselines by cutting down the number of communication rounds. This leads to faster convergence while maintaining or improving performance.

All remaining plots are depicted in Appendix D.2. We also observe a lower variance in communications for the ISP technique under Uniform sampling. This can be explained by the optimization problem

Table 3: TINY-IMAGENET setup

| Method | Communications | Test Loss | Accuracy |
|--------|----------------|-----------|----------|
| FEDCOR | 1520 | 0.8849 | 0.7945 |
| ISP FEDCOR | **1344** | **0.8787** | **0.7963** |

equation 3, in which we choose $m_{\tau+1}$ based on the overall model improvement. Despite the approximation equation 5, $N = 10$ was found to be sufficient even in the case of uniform sampling $C_{\tau+1}^n$. As noted in Problem 3, the surrogate loss $\hat{f}$ enables additional optimization in expectation estimation in terms of client inference. Its impact shown in Appendix D.4. We also discuss the relaxing of this assumption in Appendix E.8.

## 5.3 ECG EXPERIMENT

For ECG classification, we employ a federated learning setup with 2,000 clients, significantly exceeding conventional federated learning scales and providing a rigorous stress-test for our approach. Instead of the absence of the surrogate loss $\hat{f}$ in Section 5.2, here we derive it from the publicly available PTB-XL dataset (Wagner et al., 2020). The client subset size $m = 400$ was chosen following the proportion in Section 5.2, addressing the non-trivial challenge of optimal client selection in real-world deployments. Successful operation under these conditions validates exceptional robustness and readiness for large-scale industrial applications.

Since an ECG can be characterized by more than one pathology, we address the multi-label classification of the 6 well-studied (Ribeiro et al., 2020; Hannun et al., 2019) abnormalities: Atrial FIBrillation (AFIB), Premature Ventricular Complex (PVC), Complete Left (Right) Bundle Branch Block (CLBBB, CRBBB), Left Anterior Fascicular Block (LAFB) and first-degree AV Block (1AVB). A widespread specific of the medical domain is a strong imbalance toward healthy people. To address this, we integrated positive example weights in local client loss functions and used domain-specific metrics such as Sensitivity and Specificity, which are weighted averaged across pathologies (Strodthoff et al., 2020).

Table 4 and Figure 2b summarize the experimental results, demonstrating metrics comparable to state-of-the-art ones (Ribeiro et al., 2020; Hannun et al., 2019). Our approach reduces communication and time overhead by more than **67%** (compared to $m = 400$) without substantial degradation in downstream performance. This improvement stems from the discrepancy between the median number of clients actually selected by the ISP technique and the baseline choice of $m$, which highlights the method's scalability to large client populations and its practical viability for real-world deployments when the appropriate global client count is unknown. In the clinically realistic ECG task – where client participation strongly affects performance – ISP exhibited an average selection of 68 clients per round. To study this behavior, we compare against a uniform baseline that samples exactly 68 clients, with results summarized in Table 4.

Table 2: Impact of ISP technique with various client selection strategies. Accuracy and Loss statistics obtained over 5 runs. Training time represents the average duration across runs, formatted as hours, minutes, and seconds. Communication cost is defined by the number of client-server exchanges, capturing the overhead of large payload transfers from distributed nodes – this includes full intermediate $\tau + 1/2$ communication rounds.

| Method | Communications ($\downarrow$) | Test Loss ($\downarrow$) | Accuracy ($\uparrow$) | Avg Training Time ($\downarrow$) |
|--------|-------------------------------|--------------------------|------------------------|----------------------------------|
| Uniform | $17{,}767 \pm 1{,}937$ | $0.461 \pm 0.009$ | $\mathbf{0.842 \pm 0.004}$ | 8h 54m 49s |
| ISP-Uniform | $\mathbf{14{,}343 \pm 556}$ | $0.464 \pm 0.022$ | $0.836 \pm 0.013$ | **6h 14m 21s** |
| POW-D | $\mathbf{10{,}347 \pm 493}$ | $0.573 \pm 0.012$ | $0.812 \pm 0.009$ | **3h 10m 33s** |
| ISP-POW-D | $10{,}864 \pm 546$ | $0.526 \pm 0.030$ | $0.830 \pm 0.011$ | 3h 20m 14s |
| FedCor | $19{,}360 \pm 557$ | $0.449 \pm 0.017$ | $0.848 \pm 0.006$ | 5h 44m 05s |
| ISP-FedCor | $\mathbf{15{,}037 \pm 468}$ | $0.439 \pm 0.011$ | $0.848 \pm 0.010$ | **4h 34m 57s** |
| FedCBS | $19{,}207 \pm 837$ | $0.507 \pm 0.018$ | $0.830 \pm 0.007$ | 11h 18m 09s |
| ISP-FedCBS | $\mathbf{16{,}611 \pm 816}$ | $0.487 \pm 0.007$ | $0.836 \pm 0.001$ | **9h 28m 14s** |
| DELTA | $19{,}700 \pm 191$ | $0.816 \pm 0.019$ | $0.721 \pm 0.007$ | 32h 05m 21s |
| ISP-DELTA | $\mathbf{14{,}791 \pm 214}$ | $0.814 \pm 0.029$ | $0.722 \pm 0.010$ | **21h 45m 26s** |

Table 4: Multilabel ECG classification performance with ISP technique. The first comparison was made with the default number of participants ($m = 400$), while the second one with the ISP average selection rate ($m = 68$). The downstream metrics are weighted over pathologies.

| Method | Communications ($\downarrow$) | Specificity ($\uparrow$) | Sensitivity ($\uparrow$) | Training Time ($\downarrow$) |
|---|---|---|---|---|
| UNIFORM ($m = 400$) | 106400 | **0.9129** | 0.8511 | 22h 25m 50s |
| ISP UNIFORM | **34975** | 0.9049 | **0.8596** | **4h 41m 51s** |
| UNIFORM ($m = 68$) | **28560** | 0.8913 | 0.8364 | **4h 13m 54s** |
| ISP UNIFORM | 34975 | **0.9049** | **0.8596** | 4h 41m 51s |

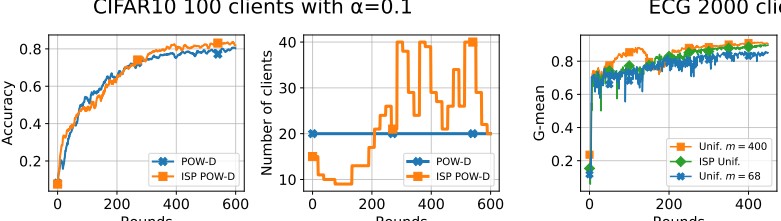

(a) Client count dynamics and test accuracy for POW-D (2nd run).

(b) Client count dynamics and test Geometric mean (G-mean) for UNIFORM on ECG.

Figure 2: (a) ISP-POW-D achieves optimal performance by round 415, POW-D's 504. (b) ISP-UNIFORM achieves optimal performance by round 416, UNIFORM with $m = 400$ and $m = 68$ clients – 266 and 420 round, respectively. For clarity, results are shown through round 600 and 450 for (a) and (b) respectively, omitting $\tau + 1/2$ ISP communications.

While a fixed-size sampling strategy set to the median may achieve slightly lower nominal communication costs than ISP, it incurs a marked degradation in model performance and cannot react to transient fluctuations in client utility. By contrast, ISP adaptively varies the number of participants over time, which helps to navigate the non-linear dependencies common in real FL tasks. These results imply that an effective median is useful but not sufficient: temporal adaptivity in participant count is needed to balance communication efficiency and convergence robustness.

## 5.4 GRADIENT COMPRESSION

The extensive results on the impact of ISP techniques under client sampling strategies (Tables 2-4) raise reasonable questions about its robustness compared to other approaches to effective communication. Modern architectures are computationally intensive, and despite the progress made with remote devices and the optimization of neural networks, the task of transmitting such a volume of local calculations remains critical and largely determines the communication bottleneck. Therefore, an important communication-efficient practical direction is the compression of transmitted information. We consider the selection of the number of clients in conjunction with QSGD (Alistarh et al., 2017) compression. We examine TOPK and RANDK gradient coordinate selection as compression methods with $K$ equal to $5\%$ and $15\%$ of the total dimension, respectively. Since compression has difficulty in non-convex convergence for heterogeneous partitioning (Sahu et al., 2021), we conduct uniform ones across clients. The results of the ISP technique along with gradient compression are presented in Table 5.

It is worth noting that the comparison is also carried out by the number of client-server communications since the volume of transmitted updates is fixed. Thus, the obtained gain demonstrates the principal possibility of integrating ISP technique with different communication-efficient approaches, while achieving up

Table 5: Gradient Compression setup

| Method | Communications | Test Loss | Accuracy |
|---|---|---|---|
| RANDK | 25040 | 0.4506 | 0.8438 |
| ISP RANDK | **17162** | **0.4447** | **0.8568** |
| TOPK | 25080 | **0.4435** | **0.8650** |
| ISP TOPK | **21436** | 0.4644 | 0.8538 |

to $30\%$ additional reduction without a significant loss of overall performance. We attribute this increase to the uniform partitioning of the dataset, which is reflected in stricter solutions of equation 3. As a consequence, the median number of clients decreases, leading to a more communication-efficient procedure.

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

## A    APPENDIX

## B    ADDITIONAL PSEUDOCODE

---

**Algorithm 3** EXPECTESTIM

---

**Require:** Client states $\{\boldsymbol{x}_i^{\tau+1/2}\}_{i=1}^M$, number of participants $m$.
    **Context:** strategy $\mathcal{S}_\tau$, loss history $\{f(\boldsymbol{x}^t)\}_{t=1}^\tau$, depth $N$.
1:  $f(\boldsymbol{x}^{\tau+1}) \leftarrow 0$
2:  **for** $n = 1$ to $N$ **do**
3:     $C_{\tau+1}^n \sim \mathcal{S}_\tau, \quad |C_{\tau+1}^n| = m$                            ▷ Sample subset
4:     $\boldsymbol{x}^{(\tau+1/2)n} \leftarrow Aggregate(\{\boldsymbol{x}_i^{\tau+1/2}\}_{i \in C_{\tau+1}^n})$
5:     **if** $\hat{f}$ available **then**                        ▷ Surroagte loss availability
6:        $f(\boldsymbol{x}^{(\tau+1)n}) \leftarrow \hat{f}(\boldsymbol{x}^{(\tau+1)n})$
7:     **else**
8:        **for all** clients $i \in C_{\tau+1}^n$ **in parallel do**
9:           $f_i(\boldsymbol{x}^{(\tau+1)n}) \leftarrow$ CLIENTINFER$(\boldsymbol{x}^{(\tau+1)n}, f_i)$
10:       **end for**
11:       $f(\boldsymbol{x}^{(\tau+1)n}) \leftarrow \sum_{i \in C} w_i f_i(\boldsymbol{x})$
12:     **end if**
13:     $f(\boldsymbol{x}^{\tau+1}) \leftarrow f(\boldsymbol{x}^{\tau+1}) + \frac{1}{N} f(\boldsymbol{x}^{(\tau+1)n})$
14: **end for**
15: $f(\boldsymbol{x}^{\tau+1}) \leftarrow EMA(\{f(\boldsymbol{x}^{\tau+1}), f(\boldsymbol{x}^\tau), \dots\})$            ▷ Smoothing
16: $\delta f_{\tau+1/2}(m) \leftarrow f(\boldsymbol{x}^{\tau+1}) - f(\boldsymbol{x}^\tau)$
17: **return** $\delta f_{\tau+1/2}(m)$

---

## C    TECHNICAL DETAILS

This section provides a comprehensive overview of the technical setup underlying our experiments. We detail the computational resources and software environment used to simulate the federated learning setting, followed by an explanation of key implementation aspects such as client-server orchestration, data partitioning, and communication handling. Finally, we summarize the hyperparameter configurations used across all datasets to ensure fair and consistent comparisons (Table 6).

**Computer resources.** Our implementation is implemented in Python 3.10 and relies on CUDA 12.8 with PYTORCH 2.6.0 for GPU acceleration. All experiments (Section 5, Appendix D, E) simulate a distributed setup on a single server, powered by an AMD EPYC 7513 32-core CPU (2.6 GHz base clock) and two NVIDIA A100 SXM4 GPUs (80 GB each) interconnected via NVLink. A detailed description of libraries, versions and code for reproducibility is presented in our codebase [2].

**Code specifics.** The code for our simulation models federated distribution by instantiating a pool of worker processes via Python multiprocessing: each client process is launched with its own pipe endpoint for bidirectional communication with the server and unique rank identifier. A central manager then orchestrates these processes by batching client launches due to the GPU memory constraints of the computer server. All clients are given identical CPU/GPU allocations, and hence the clients are system homogeneous across experiments.

Within each client, its local dataset is split stratified into training and validation subsets in a ratio of 4 by 1. This division ensures robust global model validation for accurate communication-cost estimation between experiments with a variable number of clients, and stratification preserves class proportions under heterogeneous Dirichlet sampling (except gradient compression in Section 5.4).

---

[2]https://anonymous.4open.science/r/ISPFL/

Finally, if any client has fewer than four samples of a given minority class, all such samples are retained in its training partition to avoid splitting inconsistency. We measure communication cost solely by the number of client-to-server exchanges, reflecting the primary bottleneck of transmitting large payloads from distributed nodes. Thus, the *full-intermediate* communication rounds are also taken into account when calculating communications.

**Hyperparameters.** To ensure a fair comparison, we maintain consistent hyperparameters across all methods. Experiment parameters for each dataset are provided in Table 6, while complete hyperparameter specifications (including normalization factors $\mu_1, \sigma_1, \mu_2, \sigma_2$) are available in our codebase (configs folder).

Table 6: Parameters for FL experiments across Datasets

| Parameter | CIFAR-10 | ECG | Tiny ImageNet |
|---|---|---|---|
| **Input Type** | RGB Image | 12-channel Signal | RGB Image |
| **Num Classes** | 10 (Multi-class) | 6 (Multi-label) | 200 (Multi-class) |
| **Model** | ResNet18 | ResNet1D18 | Swin-T |
| **Loss Function** | CrossEntropyLoss | BCEWithLogitsLoss | CrossEntropyLoss |
| **Optimizer** | Adam SGD (DELTA) | Adam | AdamW |
| **Learning Rate** | Adam: 0.003 SGD (DELTA): 0.003 | 0.003 | 0.001 |
| **Weight Decay** | Adam: 0 SGD (DELTA): 0 | 0 | 0.01 |
| **Batch Size** | 64 | 128 | 256 |
| **Preprocessing** | $\mathcal{N}(\mu_1, \sigma_1)$ | $\mathcal{N}(0, 1)$ | $\mathcal{N}(\mu_2, \sigma_2)$ |
| **Augmentations** | RandomCrop ($32 \times 32$) HorizontalFlip | No | No |

# D EXPERIMENTS DETAILS

## D.1 ADDITIONAL EXPERIMENTS

### D.1.1 TEXT DOMAIN

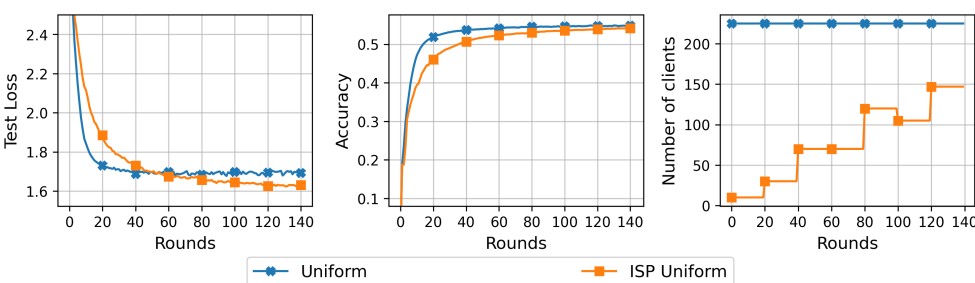

Figure 3: Test loss, accuracy, and client count dynamics, for the ISP technique under UNIFORM client sampling on SHAKESPEARE dataset. For clarity, results are shown through round 140, omitting $\tau + 1/2$ ISP communications. ISP-UNIFORM achieves optimal performance by round 132, UNIFORM's by 80.

To verify the robustness of the ISP technique, we additionally evaluated it for another domain of language–modeling task using the SHAKESPEARE dataset from the LEAF benchmark (Caldas et al., 2018). Following LEAF, each speaking role in the plays is treated as an individual participant, yielding $M = 1129$ clients with highly non-IID data. The downstream task is next–character prediction: each training sample is a sequence of $80$ consecutive characters extracted from the original plays, and the target is the next character in the stream. We preserve the natural temporal order of each play and split every client's stream into train and test parts in a $9/1$ ratio along time, so that future characters do not leak into the training set. Each character is embedded into an $8$-dimensional vector and processed by a two-layer LSTM with $256$ hidden units per layer as in the LEAF benchmark. The learning rate of ADAM optimizer is equal to $0.01$.

We evaluated the ISP technique compared to UNIFORM client sampling with fixed cohort of $m = 225$ clients per round, following the participation ratio of the baseline CIFAR-10 experiment (Section D.3). We run intermediate ISP communication for each $\Delta = 20$ rounds with coverage depth $N = 10$ and resolution $w = 20$ to solve equation 4. We are relaxing full-client intermediate communication to half of all clients $m_{\tau+1/2} = 550$. The momentum parameter was fixed to $\beta = 0.5$. To measure the performance of the global model, we report loss on the holdout test split, TOP-1 character prediction accuracy, and perplexity computed from the test loss. Table 7 and Figure 3 demonstrate the results of the experiment.

The same as in Section 5.3, we have significant communication savings due to the difference in the average number of participants for the constant and dynamic strategies. The ISP technique averaged 81 clients per round of communication and doesn't degrade on the downstream task, slightly reducing TOP-1 Accuracy, but improving in Test Loss and, as a result, in Perplexity score. In Figure 3 we see a clear increase in the number of participating clients, demonstrating the adaptability of ISP as it goes to the optimum.

Table 7: Comparison of ISP on the SHAKESPEARE next-character prediction under UNIFORM client sampling.

| Method | Communications (↓) | Test Loss (↓) | Accuracy (↑) | Perplexity (↓) |
|---|---|---|---|---|
| UNIFORM | 18000 | 1.676 | **0.546** | 5.35 |
| ISP-UNIFORM | **13015** | **1.623** | 0.541 | **5.07** |

### D.1.2 OORT EXAMPLE

To address the diversity of client sampling methods, we further evaluate ISP on the CIFAR-10 classification task under the OORT (Lai et al., 2021) strategy baseline. In Section 2 we discussed that existing client selection methods can be coarsely grouped by whether they primarily target statistical or system heterogeneity. In contrast to the other baselines considered in Section 5.2, OORT explicitly couples both aspects in a single utility score. Therefore, this experiment not only expands the set of baselines but also probes the robustness of ISP when client utilities depend simultaneously on data value and runtime.

For each client $i$, OORT defines a utility

$$\text{Util}(i) = \underbrace{|B_i| \sqrt{\frac{1}{|B_i|} \sum_{k \in B_i} \text{Loss}(k)^2}}_{\text{statistical utility } U(i)} \cdot \underbrace{\left(\frac{T}{t_i}\right)^{\mathbf{1}_{\{T < t_i\}} \cdot \alpha}}_{\text{global system utility}},$$

where $B_i$ is the set of local mini-batches used to estimate the client's contribution, $\text{Loss}(k)$ is the training loss on batch $k$, $t_i$ is the profiled completion time of client $i$, $T$ is a developer-preferred duration of each communication round, and $\alpha$ is a penalty factor. The first term $U(i)$ favors clients with both large and diverse local datasets, while the second term penalizes clients whose execution time exceeds $T$ with exponential penalty factor $\alpha$.

We run the OORT baseline with exploration factor $\varepsilon = 0.05$. At each round the server selects the top $(1 - \varepsilon) \times m$ participants by the utility $\text{Util}(i)$ and then explores additional clients whose utilities lie within a confidence interval with chosen width $c = 0.6$. This configuration allows OORT to systematically exploit high-utility clients while still visiting most of all participants during the course of training. On top of this sampling process we deploy ISP technique with the baseline parameters, considering for Image domain (see Table 1 in the main part for details).

In our code base, all clients share identical hardware resources. Nevertheless, system heterogeneity naturally arises because of non-i.i.d. data client distribution: with a fixed batch size, clients with larger local datasets perform more training steps per local epoch and therefore have longer execution times. As a result, in our implementation, the runtime $t_i$ is directly proportional to the amount of local data, and the system component of $\text{Util}(i)$ implicitly reduces the value of clients with more data. To eliminate this contradiction, we inject artificial delays. We first sample a subset of $p_1 = 0.1$ clients. Whenever any of these clients participates in a round, we draw a random $\xi \sim \text{Geom}(p_2)$ number of delayed batches and add $0.5 \cdot \xi$ seconds of sleep time to their local training, where each $0.5$ seconds corresponds to the measured duration of one training step. We chose $p_2 = 0.1$ and the exponential factor $\alpha = 1$ for a reasonable regularization in client utility.

We choose a conservative estimate of developer-preferred duration $T$. We profile the federated training loop, excluding technical computations such as global evaluation on the validation subset and assuming a batch size of $64$ for all CIFAR-10 experiments (see Table 6), one training step takes $\approx 0.5$ seconds. With 50,000 training examples and $100$ participating clients, we treat 1,500 local examples as a reasonable upper bound per client, which corresponds to roughly $24$ training steps, and with additional communication and bookkeeping overhead, this yields a developer-preferred round duration of $T \approx 14$ seconds. Thus, $T$ specifies the maximum acceptable waiting time, after which the system term begins to penalize slow clients.

Table 8 and Figure 4 summarize the results. Empirically, ISP-OORT reduces communication overhead by slightly more than 20% compared to standard OORT, while preserving final accuracy and substantially lowering the test loss. This improvement indicates that ISP can enhance the efficiency of client-selection methods even when integrated with existing system-aware approaches. These

Table 8: Comparison of ISP on the CIFAR-10 classification under OORT client sampling strategy.

| Method | Communications ($\downarrow$) | Test Loss ($\downarrow$) | Accuracy ($\uparrow$) |
|---|---|---|---|
| OORT | 19500 | 0.478 | 0.834 |
| ISP-OORT | **15448** | **0.454** | 0.835 |

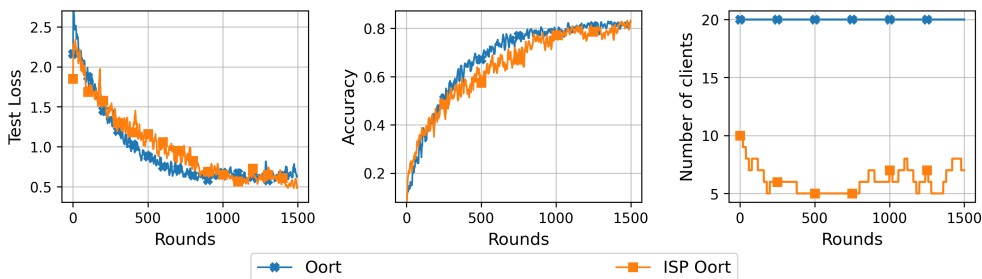

Figure 4: Test loss, accuracy, and client count dynamics, for the ISP technique under OORT client sampling strategy on CIFAR-10 dataset. For clarity, results are shown through round 1500, omitting $\tau + 1/2$ ISP communications. ISP-OORT achieves optimal performance by round 1334, while OORT's by 975.

results demonstrate the feasibility of applying our method in scenarios characterized by system heterogeneity.

### D.1.3 ADAFL COMPARISON

We continue to develop the results of the Table 11 by addressing to the ISP dynamic of $m_\tau$. As we see from Figure 5, ISP-UNIFORM has a clear non-linear dynamic with growth of $m_\tau$ at the end, which is consistent with the ADAFL (Li et al., 2024) mentioned in Related Work (see Section 2). However, the proposed technique represents a linear schedule with a positive trend; as we approach the optimum, we become more sensitive to the heterogeneity of local updates. The original work exploits the schedule from the initial $m_0 = 10$ to the *full*-round communication $m_T = 100$ in a close CIFAR-10 setup. However, ADAFL does not directly address communication efficiency and, as we have seen from the results in Table 10, a significant increase in the number of participants does not provide the same gain in downstream performance that it takes away for communication exchanges. To address the communication bottleneck in addition to the baseline schedule, we will consider a communication-quality trade-off zone ranging clients from $m = 5$ to $m = 20$ every 50 rounds and then reaching a plateau. We call these schedules *full* and *econom*, respectively. Table 9 summarizes the results of the comparison.

Table 9: Comparison of ISP with linear scheduler dynamics.

| Method | Communications | Test Loss | Accuracy |
|---|---|---|---|
| UNIFORM | 17,767 | 0.461 | 0.842 |
| ADAFL (ECONOM) | 16,191 | 0.508 | 0.827 |
| ADAFL (FULL) | 63,876 | **0.421** | **0.855** |
| ISP-UNIFORM | **14,343** | 0.464 | 0.836 |

We observe that the linear increasing *econom* scheduler does not achieve the same efficiency as the ISP technique. This demonstrates the need for an intelligent approach to adaptive choice of the number of clients. At the same time, there is a significant spread depending on the chosen schedule scenario and ADAFL does not relieve us from the choice of the boundaries $m_0$ and $m_T$ and the scheduler step, which also significantly affect the federated learning process, as well as the number of clients in the current round.

This section presents a thorough account of our experimental evaluation. In Section D.2, we show additional plots of test loss, accuracy, and client-count dynamics (see Figures 5–6). Section D.3 analyzes baseline participant counts under uniform sampling and the resulting communication–quality trade-offs (Table 10), while Section D.4 reports computation-time measurements for ISP intermediate communications (Table 12 and Table 13). Finally, Section D.1.3 compares our adaptive ISP strategy against the linear schedules of AdaFL (Table 9), highlighting the benefits of intelligent client selection.

### D.2 Additional Plots

In this section, we present and discuss test loss, accuracy, and client count dynamics for the remaining experiments in the main part (Section 5). Figures 5-6 illustrate them. Because we performed the experiments five times for Table 2, the corresponding Figure 5 illustrates the second run, which was selected randomly.

Figure 5: Test loss, accuracy, and client count dynamics, for the ISP technique under various client selection strategies (2nd run). For clarity, results are shown through round 1500, omitting $\tau + 1/2$ ISP communications.

As can be seen from the client count dynamics in Figure 5, the ISP technique considers between $m = 5$ and $m = 10$ clients throughout the training for all client strategies except FEDCBS (and POW which we discussed in Section 5.2). This is consistent with the results of Appendix D.3 (Table 10) in which the described range of clients leads to communication savings of up to $30\%$ for UNIFORM sampling. However, in contrast to the baseline with fixed $m$, ISP preserves downstream performance for all client strategies, which is the result of intelligent selection based on the target optimization of equation 3 and we address them in detail in Appendix D.3, D.1.3.

This is also reflected in the convergence speed for DELTA and FEDCOR client strategies, since for them the choice of $m_{\tau+1}$ is closer to $m = 5$. Despite the stochastics and slower convergence, the intelligent selection of the number of participants is based on the improvement of the global model, so we observe stable convergence to comparable quality indicators.

The results in Figure 6 show the same trends as when adding ISP to client sampling methods, except for the optimal range $m_{\tau+1}$. This again highlights the non-trivial nature of choosing $m$ even for a fixed schedule and demonstrates the practical importance of intelligence selecting the number of participants.

### D.3 Baseline number of participants

To correctly determine the number of participants $m$ in communication rounds for non-adaptive baselines, we performed an extensive comparison of it for the UNIFORM client selection strategy

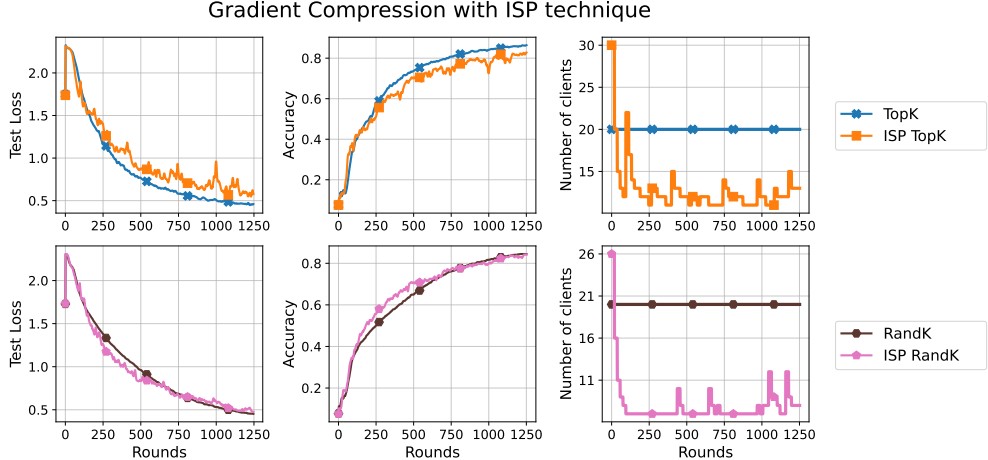

Figure 6: Test loss, accuracy, and client count dynamics, for the ISP technique under various gradient compression methods. We set $K$ equal to $5\%$ and $15\%$ of the total dimension for the TOPK and RANDK compressions, respectively. For clarity, results are shown through round 1250, omitting $\tau + 1/2$ ISP communications.

under the same conditions as described in Section 5.1 for Table 2. For $m = 20$, we take the average results from the main part of the article (see Table 2).

Table 10: The number of participants $m$ for downstream CIFAR-10 task with UNIFORM client selection strategy.

| Number of clients | Communications | Test Loss | Accuracy |
|---|---|---|---|
| $m = 5$ | 12,065 | 0.624 | 0.791 |
| $m = 10$ | 15,830 | 0.536 | 0.825 |
| $m = 15$ | 18,315 | 0.473 | 0.838 |
| $m = 20$ | 17,767 | 0.461 | 0.842 |
| $m = 25$ | 30,350 | 0.458 | 0.842 |
| $m = 40$ | 46,720 | 0.468 | 0.840 |
| $m = 50$ | 62,250 | 0.474 | 0.836 |

Table 10 summarizes this comparison and provides several insights.

- **Diminish returns beyond** $m = 20$: We see a clear non-optimal amount of communication costs for $m > 20$ to obtain the same level of target performance as for $m = 20$. This indicates the well-known statement about the suboptimal nature of a large number of participants in a setup with strong heterogeneity of local client data.

- **Communication-Quality trade-off:** As we move to $m < 20$ we cannot say for sure which of the values $m$ is optimal in terms of communication-quality performance. At $m = 10$ we see an acceptable compromise: $11\%$ reduction in communication exchanges while degrading the final accuracy by $1.7\%$. This is especially true in the case where communication efficiency plays a decisive role.

Despite the communication efficiency at $m = 10$, we preferred to choose $m = 20$, focusing on the best final target quality for ease of comparison: while the ISP technique demonstrates a significant reduction in communication costs, downstream performance remains at the same level. In the case of $m = 10$, we have the described trade-off, in which excessive communication savings lead to accuracy degradation, which requires additional mention in the main part of the work. Table 11 encapsulates the appropriate numbers of participants $m_\tau$, which is discussed above with an adaptive ISP strategy for comparison. Despite the communication-quality trade-off, ISP demonstrates clear

Table 11: Comparison of ISP technique against most appropriate number of participants $m$ for CIFAR-10 with UNIFORM sampling.

| Number of clients | Communications | Test Loss | Accuracy |
|---|---|---|---|
| $m = 10$ | 15,830 | 0.536 | 0.825 |
| $m = 20$ | 17,767 | **0.461** | **0.842** |
| ISP $m_\tau$ | **14,343** | 0.464 | 0.836 |

superiority in communication savings and maintains downstream performance, which supports the similar results of Table 4 in the ECG domain.

### D.4 COMPUTATION TIME

The measurement of time was performed up to the best training round, which was identified based on the loss observed on the clients' validation datasets. Upon receiving the global model, each client first conducted validation before commencing local training. Owing to the implementation details of our framework (see Appendix C), full parallelism in client training cannot be simulated. As a result, in all experiments, clients are organized into batches. Within each batch, clients are trained concurrently, and training proceeds sequentially from one batch to the next, thereby introducing a controlled degree of sequential execution into the experiments. This limitation is particularly consequential when the number of participating clients varies. Under a fully parallel regime, the duration of a communication round is invariant with respect to the number of clients. In contrast, our implementation processes clients in fixed-size batches, and the total number of batches increases proportionally with the number of participants per round. Consequently, the measured duration of a communication round exhibits an explicit dependence on the number of clients involved.

Table 12 reports the computational overhead introduced by the ISP technique as a result of intermediate communication (see Problem 1). We present the average overhead of full-client intermediate communication and quantify its relative contribution in comparison to a standard training round, given that ISP is positioned as a strategy with communication triggered every $\Delta = 20$ rounds. We see that overall overhead is no more than $10\%$ which is neglect due to communication savings.

Table 13 reports the computational overhead introduced by the ISP technique under a different relaxing strategies: increasing step size, adding surrogate loss, inreasing $\Delta$. With this strategies we can almost neglect computational overhead in overall training.

Table 12: Computational overhead for ISP intermediate communication and overal FL training.

| Method | Intermediate Computational Overhead | Overall Overhead |
|---|---|---|
| ISP-UNIFORM | $\times 1.633$ | $\times 1.032$ |
| ISP-FEDCBS | $\times 2.244$ | $\times 1.062$ |
| ISP-DELTA | $\times 1.443$ | $\times 1.026$ |
| ISP-POW | $\times 3.380$ | $\times 1.093$ |
| ISP-FEDCOR | $\times 1.308$ | $\times 1.018$ |

Table 13: Computational overhead for ISP intermediate communication and overal FL training under sparsing intermediate communication.

| Method | Intermediate Computational Overhead | Overall Overhead |
|---|---|---|
| ISP-UNIFORM | $\times 1.633$ | $\times 1.032$ |
| ISP-UNIFORM with $\hat{f}$ | $\times 1.144$ | $\times 1.006$ |
| ISP-UNIFORM with $w = 3$ | $\times 1.213$ | $\times 1.021$ |
| ISP-UNIFORM with $\Delta = 100$ | $\times 1.633$ | $\times 1.013$ |

## D.5 COMMUNICATION ANALYSIS

Table 14 summarizes the communication savings and overheads of the ISP technique across all domains and experiments considered. Despite the noticeable overhead of the ISP method for certain experiments, it consistently provides communication savings compared to the baselines. We address the reduction of this overhead in Table 15, providing optimization of up to 7% of the total number of communications.

Table 14: Communication efficiency and overhead for ISP technique

| Domain | Method | Overall Communications | Intermediate Communications | Overall Overhead | Overall Savings |
|---|---|---|---|---|---|
| IMAGE | UNIFORM | $17,767 \pm 1,937$ | $0 \pm 0$ | $\times 1.00 \pm 0.00$ | $0\% \pm 0\%$ |
| | ISP-UNIFORM | $14,343 \pm 556$ | $6,200 \pm 310$ | $\times 1.43 \pm 0.03$ | $-19\% \pm 9\%$ |
| | POW | $10,347 \pm 493$ | $0 \pm 0$ | $\times 1.00 \pm 0.00$ | $0\% \pm 0\%$ |
| | ISP-POW | $10,864 \pm 546$ | $2,100 \pm 105$ | $\times 1.19 \pm 0.01$ | $+5\% \pm 6\%$ |
| | FEDCOR | $19,360 \pm 557$ | $0 \pm 0$ | $\times 1.00 \pm 0.00$ | $0\% \pm 0\%$ |
| | ISP-FEDCOR | $15,037 \pm 468$ | $7,700 \pm 385$ | $\times 1.51 \pm 0.03$ | $-22\% \pm 4\%$ |
| | FEDCBS | $19,207 \pm 837$ | $0 \pm 0$ | $\times 1.00 \pm 0.00$ | $0\% \pm 0\%$ |
| | ISP-FEDCBS | $16,611 \pm 816$ | $6,700 \pm 335$ | $\times 1.40 \pm 0.03$ | $-14\% \pm 6\%$ |
| | DELTA | $19,700 \pm 191$ | $0 \pm 0$ | $\times 1.00 \pm 0.00$ | $0\% \pm 0\%$ |
| | ISP-DELTA | $14,791 \pm 214$ | $7,300 \pm 365$ | $\times 1.49 \pm 0.03$ | $-24\% \pm 2\%$ |
| | OORT | $19,500$ | $0$ | $\times 1.00$ | $0\%$ |
| | ISP-OORT | $15,488$ | $6,600$ | $\times 1.30$ | $-21\%$ |
| | RANDK | $25,040$ | $0$ | $\times 1.00$ | $0\%$ |
| | ISP-RANDK | $17,162$ | $6,200$ | $\times 1.36$ | $-31\%$ |
| | TOPK | $25,080$ | $0$ | $\times 1.00$ | $0\%$ |
| | ISP-TOPK | $21,436$ | $6,200$ | $\times 1.29$ | $-15\%$ |
| | IMAGENET | $1,520$ | $0$ | $\times 1.00$ | $0\%$ |
| | ISP-IMAGENET | $1,344$ | $300$ | $\times 1.22$ | $-12\%$ |
| ECG | UNIFORM | $106,400$ | $0$ | $\times 1.00$ | $0\%$ |
| | ISP-UNIFORM | $34,975$ | $8000$ | $\times 1.24$ | $-67\%$ |
| TEXT | UNIFORM | $18,000$ | $0$ | $\times 1.00$ | $0\%$ |
| | ISP-UNIFORM | $13,015$ | $3850$ | $\times 1.30$ | $-28\%$ |

Estimating future loss changes $\delta f_\tau(m_{\tau+1})$ requires ISP method adds intermediate client participation (see Algorithm 2, Line 4), which affects overall communication efficiency and introduces a corresponding overhead. This overhead is largely determined by the intermediate frequency $\Delta$, which acts as a knob that trades off adaptivity for extra communication, and the number of corresponding communicating clients $m_{\tau+1/2}$. Relaxing any of them steadily lowers the overhead while keeping test loss and accuracy essentially unchanged. In the edge case, we can run intermediate communication once to carefully determine the number of participants compared to guessing, which largely determines the communication efficiency in experiments with Text and ECG domains.

Table 15: Communication overhead for ISP intermediate communication across relaxed hyperparameters of this procedure on CIFAR-10 dataset under UNIFORM client sampling.

| Method | Communications | Overhead | Test Loss | Accuracy |
|---|---|---|---|---|
| ISP-UNIFORM | 14,343 | $\times 1.430$ | 0.464 | 0.836 |
| ISP $m_{\tau+1/2} = 80$ | 12,842 | $\times 1.311$ | 0.461 | 0.840 |
| ISP $m_{\tau+1/2} = 60$ | 15,900 | $\times 1.252$ | 0.467 | 0.845 |
| ISP $\Delta = 100$ | 12,652 | $\times 1.089$ | 0.472 | 0.831 |
| ISP $\Delta = 100, m_{\tau+1/2} = 80$ | 12,924 | $\times \mathbf{1.069}$ | 0.465 | 0.837 |

# E ABLATION STUDY

In this section, we provide an extensive ablation study of the main components of the ISP technique. We discussed them in detail in the main part of Problems 1-3. The goal of this ablation is to systematically adjust these parameters compared to those chosen in the main part of the experiments (Table 1) to better understand their impact on communication-efficiency caused by the optimization of equation 3. By tuning these parameters, we also aim to identify which ones are more or less sensitive (see Appendix E.1-E.4). For hyperparameters that directly affect the optimization time equation 3 (see Appendix E.5, E.6) we perform a comparison of the intermediate round communication times similar to the measurements in Appendix D.4. In Appendix E.7 we ablate the optimization problem equation 3 itself, while in Appendix E.8 we address the relaxation of the surrogate loss function.

## E.1 INTERMEDIATE PARTICIPANTS NUMBER

Practical-oriented federated learning scenarios often feature partial client participation due to connectivity or resource constraints (McMahan et al., 2017; Li et al., 2020b). Addressing the limitation in Problem 1, we relax the assumption of universal client availability during intermediate synchronization ($m_{\tau+1/2} = M$). This modification samples subsets of available clients $C_{\tau+1/2} \sim \mathcal{S}_\tau$, $|C_{\tau+1/2}| < M$ rather than requiring full participation (Line 4, Algorithm 2). This relaxation immediately limits the approximation equation 5, since modeling all possible global states of $x^{(\tau+1/2)_n}$ has become infeasible. However, as we discussed in Section 5.2, even for UNIFORM sampling, $N = 10$ was found to be sufficient to cover $\delta f_{\tau+1/2}(m_{\tau+1})$ with the Monte Carlo approach. This claim is also demonstrated in the Appendix E.5, in which we explicitly ablate $N$. These considerations give us optimism that the relaxation of full intermediate communication will not lead to a dramatic degradation of the correct estimation of $m_{\tau+1}$.

We empirically evaluate this relaxation using the experimental setup shown in Table 2 under the UNIFORM sampling strategy as the one with the largest variance under statistical heterogeneity of clients for the limiting case of the approximation equation 5. The results are presented in Table 16 and Figure 7. From this Figure, we see an interesting dependency: The smaller the number of participants in intermediate communication $m_{\tau+1/2}$ the higher their number $m_{\tau+1}$ participates (except for $m_{\tau+1/2} = 20$) in the following $\Delta$ rounds according to the ISP technique. This leads to a predictable increase in communication costs, which in turn increases downstream performance. However, even reducing the full-intermediate round by 5 times ($m_{\tau+1/2} = 20$) demonstrates superiority both in communication savings and in the final Accuracy in comparison with the baseline. Interestingly, the choice of $m_{\tau+1/2} = 80$ showed a reduction in the number of communications compared to the full round strategy while slightly improving the final quality. We attribute this to the experimental error, which demonstrates the fundamental absence of deterioration of the approximation equation 5 when relaxing the condition to this choice.

This approach also handles Problem 3 when the surrogate $\hat{f}$ is unavailable. The ability to partially select clients in intermediate communication allows us to set restrictions on them. In terms of the computational efficiency of client inference in the absence of $\hat{f}$, such restrictions can be system or network requirements. This also limits the variability of the resulting $x^{(\tau+1/2)_n}$, which affects the coverage of $\delta f_{\tau+1/2}(m_{\tau+1})$, but this approach allows us to consider the problem of the absence of $\hat{f}$ as one of the restrictions along with the selection of hyperparameters $N$, $\Delta$, and resolution $w$, which is not significant in the practical implementation of ISP technique. We do not provide a relative comparison of the intermediate communication time, since ISP-UNIFORM did not choose $m_{\tau+1} > 20$ and the ablation of this hyperparameter does not directly affect the amount of computation during optimization equation 3.

Table 16: Intermediate Participants Number $m_{\tau+1/2}$ Ablation.

| Method | Communications | Test Loss | Accuracy | Overhead |
|---|---|---|---|---|
| UNIFORM | 17,766 | **0.4605** | 0.8416 | $\times 1.000$ |
| ISP $m_{\tau+1/2} = 100$ | 14,342 | 0.4635 | 0.8358 | $\times 1.432$ |
| ISP $m_{\tau+1/2} = 80$ | **12,842** | 0.4613 | 0.8403 | $\times 1.311$ |
| ISP $m_{\tau+1/2} = 60$ | 15,900 | 0.4666 | 0.8446 | $\times 1.252$ |
| ISP $m_{\tau+1/2} = 40$ | 16,873 | 0.4947 | 0.8340 | $\times 1.099$ |
| ISP $m_{\tau+1/2} = 20$ | 17,319 | 0.4673 | **0.8488** | $\times 1.059$ |

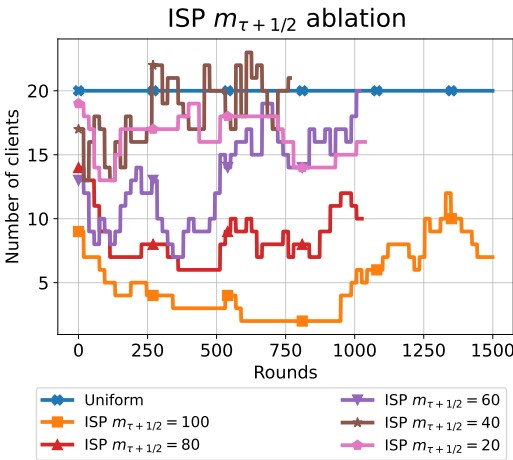

Figure 7: Client count dynamics for the ISP technique under various $m_{\tau+1/2}$ with UNIFORM sampling strategy. Results are shown up to the best round of each experiment run, as determined by client validation, omitting $\tau + 1/2$ ISP communications.

### E.2 INTERMEDIATE COMMUNICATION FREQUENCY

Another important hyperparameter of the ISP technique defined in Problem 1 is $\Delta$, which specifies the frequency of solving the problem equation 3 on the intermediate communication (see Algorithm 2). Since collecting updates from all clients incurs substantial computational and communication overhead, minimizing optimization frequency reduces client burden and system costs. This ablation study examines the degradation of downstream performance as $\Delta$ decreases compared to baseline choice of $\Delta = 20$ in experimental setup from Section 5.2. We perform ablation under the UNIFORM client selection strategy in the same settings as in the Appendix E.1 for consistency of considerations. Table 17 and Figure 8 presents the result.

We see that increasing the frequency $\Delta$ by five times ensured communication savings of up to $12\%$ compared to $\Delta = 20$ and up to $28\%$ compared to the baseline, only slightly reducing downstream performance. The subsequent increase in $\Delta$ by another 2.5 times additionally saved $13\%$ of communications with another slight decrease in accuracy. These results demonstrate the compromise Communication-Quality tradeoff described in Appendix D.3 and provide up to **38**$\%$ communication efficiency compared to the baseline. However, further rarefaction of the frequency of solving equation 3 to $\Delta = 500$ leads to a significant degradation of performance. We associate this with the blurring of the intelligent dynamics of the adaptive number of participants (see Table 11, 9), which helps to maintain target performance with communication efficiency due to the type of optimization task equation 3. However, the relaxation performed demonstrates a significant gap in the assumptions of this task (see Problems 1-3) and in the case of $\Delta = 250$ with the same Mean Relative comparison (Table 12) the total computational losses caused by the use of the ISP technique are less than **1.5**$\%$.

Table 17: Intermediate communication frequency $\Delta$ ablation.

| Method | Communications | Test Loss | Accuracy |
|--------|---------------|-----------|----------|
| UNIFORM | 17,767 | **0.461** | **0.842** |
| ISP-UNIFORM $\Delta = 20$ | 14,343 | 0.464 | 0.836 |
| ISP-UNIFORM $\Delta = 100$ | 12,652 | 0.472 | 0.831 |
| ISP-UNIFORM $\Delta = 250$ | 10,952 | 0.473 | 0.827 |
| ISP-UNIFORM $\Delta = 500$ | **9,602** | 0.480 | 0.818 |

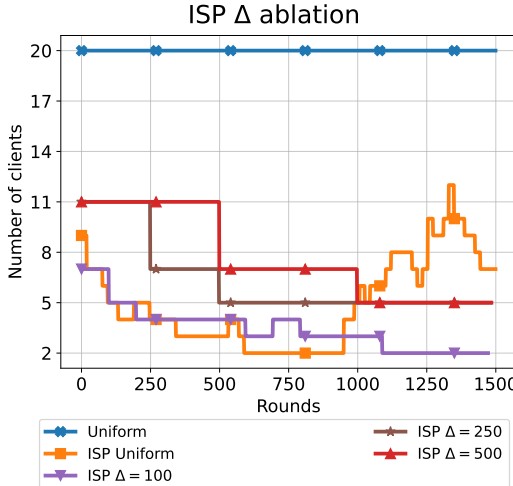

Figure 8: Client count dynamics for the ISP technique under various $\Delta$ with UNIFORM sampling strategy. For clarity, results are shown through round 1500, omitting $\tau + 1/2$ ISP communications.

### E.3 MOMENTUM $\beta$

The hyperparameter $\beta$ addresses the greedy nature of problem equation 3 by adjusting the contribution of the resulting $m_{\tau+1}$ compared to the previous history (see Line 8, Algorithm 3). We measure the impact of $\beta$ by disabling it ($\beta = 1$) in the same setup setting (see Section 5.2 and Table 1) with the UNIFORM client strategy. The results are presented in Table 18 and the summary Figure 9, which combines the dynamics of the client count for Appendix E.3 and E.4.

We see a more stochastic dynamics of the number of selected clients from Figure 9, which is reflected in a partial loss of communication efficiency without a global improvement in performance (see Table 18). Thus, the $\beta$-momentum in the updated $m_{\tau+1}$ provides greater stability to the greedy optimization in equation 3.

Table 18: Momentum $\beta$ ablation.

| Method | Communications | Test Loss | Accuracy |
|--------|---------------|-----------|----------|
| UNIFORM | 17,767 | 0.461 | 0.842 |
| ISP UNIFORM $\beta = 0.5$ | 14,343 | 0.464 | 0.836 |
| ISP UNIFORM $\beta = 1$ | 15,786 | 0.462 | 0.834 |

### E.4 EXPONENTIAL MOVING AVERAGE

We employ Exponential Moving Average (EMA) with window size $w_{ema} = 5$ to smooth global loss histories. Figure 5 shows noticeable noise in the test loss due to significant statistical heterogeneity in local clients data. This can potentially compromise loss-based optimization equation 3. This study ablates EMA's impact on the ISP procedure. Table 19 and Figure 9 encapsulate the overall results of history ablation (Appendices E.3, E.4).

Without EMA smoothing, we observe more unstable client selection patterns (see Figure 9). This stochasticity increases communication costs beyond the UNIFORM baseline despite comparable convergence. To measure the impact of the greedy optimization solving 3, we conduct an additional experiment that disables both EMA smoothing and participant momentum ($\beta = 1$), eliminating historical context. The obtained results continue the intuition about the importance of the previous context: we see a more unstable dynamic of client selection, which further leads to communication overhead.

Table 19: History usage ablation.

| Method | Communications | Test Loss | Accuracy |
|---|---|---|---|
| UNIFORM | 17,767 | 0.461 | 0.842 |
| ISP UNIFORM | 14,343 | 0.464 | 0.836 |
| ISP UNIFORM NO EMA | 24,445 | 0.466 | 0.839 |
| ISP UNIFORM NO EMA, $\beta = 1$ | 26,905 | 0.469 | 0.822 |

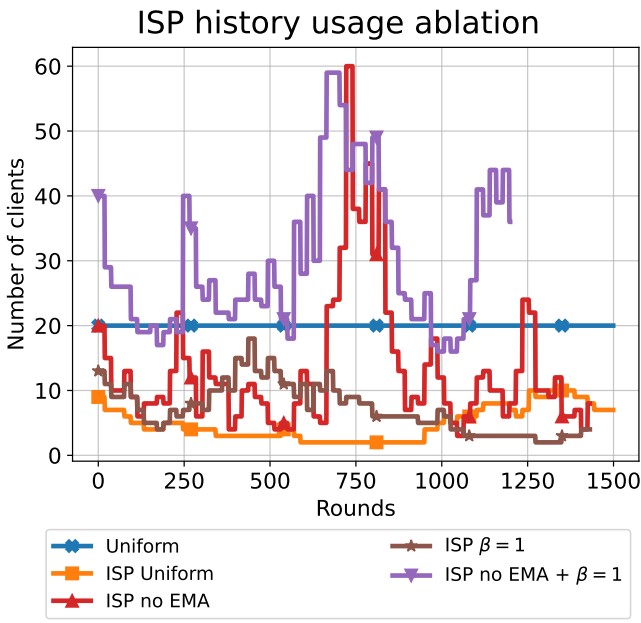

Figure 9: Client count dynamics for the ISP technique with various history usage technique with UNIFORM sampling strategy. Results are shown up to the best round of each experiment run, as determined by customer validation, omitting $\tau + 1/2$ ISP communications.

## E.5 COVERAGE DEPTH

Let us move on to the ablation of the coverage depth $N$, which is defined in Problem 2. As we noted in the discussion of Table 2, the baseline choice of $N = 10$ may have been excess, since even for the UNIFORM client sampling strategy, we observed a reduction in variance across 5 experimental runs. This choice could have a negative impact on the final computational performance (see Table 12), since the coverage depth directly determines the number of client inferences and the complexity of the intermediate communication. The purpose of this ablation is to determine the necessary but sufficient value $N$ of the coverage depth.

Table 20: Coverage depth $N$ ablation

| Method | Communications | Test Loss | Accuracy |
|--------|----------------|-----------|----------|
| UNIFORM | 17,767 | 0.461 | **0.842** |
| ISP-UNIFORM $N = 10$ | 14,343 | 0.464 | 0.836 |
| ISP-UNIFORM $N = 7$ | 14,365 | **0.453** | **0.842** |
| ISP-UNIFORM $N = 5$ | 14,290 | 0.462 | 0.839 |
| ISP-UNIFORM $N = 3$ | **14,102** | 0.476 | 0.838 |
| ISP-UNIFORM $N = 1$ | 15,102 | 0.872 | 0.696 |

Table 20 and Figure 10 present the results of the coverage depth ablation. The table confirms the assumption of redundancy of the choice of $N = 10$ even in the case of a uniform sampling strategy. We see that a gradual reduction of the coverage depth to $N = 3$ does not significantly affect the change in the results. However, for $N = 1$ we see a non-obvious behavior, in which it follows from Figure 10 that the ISP technique predominantly selects only one participant, which leads to a dramatic drop in the downstream performance. We address this to expectation estimation (see Algorithm 3), which relies on the Monte Carlo approach. Then we enumerate through $m_{\tau+1}$ beginning with $m_{\tau+1} = 1$, obtain a random $\boldsymbol{x}^{(\tau+1/2)i}$, which in the case of one selected participant $i$ is equivalent to the local client update $\boldsymbol{x}_i^{\tau+1/2}$, and estimate $f(\boldsymbol{x}_i^{\tau+1/2})$. This update can be unbiased and demonstrate model improvement in terms of $f$, and hence $f(\boldsymbol{x}_i^{\tau+1/2}) < 0$. Then, according to Algorithm 2, we will stop at $m_{\tau+1} = 1$, which may not reflect the global needs for FL convergence. In contrast, the choice $N \neq 1$ with high probability under the strong statistical heterogeneity samples the client with a biased update $\boldsymbol{x}_j^{\tau+1/2}$, which dramatically affects the overall estimate of the left hand side term in equation 5.

Table 21 shows the mean relative time comparison (similar to Table 12) as $N$ decreases. From Table 20, we have determined an appropriate sufficient coverage depth $N = 3$ that optimizes the number of computations of the solution equation 3, making them close to the original round of communication. Thus, with decreasing $\Delta$ frequency, the total computational losses becomes statistically insignificant. This shows a significant gap in Problems 1-3 in practice.

Table 21: Time measurements of ISP intermediate communications in relative to a baseline round across various coverage depth $N$.

| Method | Mean Relative Comparison |
|--------|--------------------------|
| ISP-UNIFORM $N = 10$ | 3.933 |
| ISP-UNIFORM $N = 7$ | 2.753 |
| ISP-UNIFORM $N = 5$ | 1.966 |
| ISP-UNIFORM $N = 3$ | 1.180 |
| ISP-UNIFORM $N = 1$ | 0.393 |

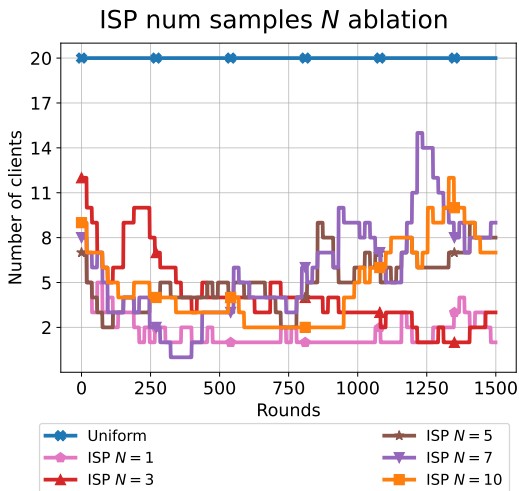

Figure 10: Client count dynamics for the ISP technique under various $N$ with UNIFORM sampling strategy. For clarity, results are shown through round 1500, omitting $\tau + 1/2$ ISP communications.

### E.6 ENUMERATING RESOLUTION

Another hyperparameter that directly affects the internal computation of optimization equation 3 (see Problem 3) is the resolution $w$, which specifies the enumeration step. This step generates a window $2w$ within which the ISP technique is not sensitive to the selected $m_{\tau+1}$. However, the results of the time measurements in Table 12 demonstrate a direct relationship between the value of $m_{\tau+1}$ and the time of the intermediate communication. Increasing the resolution window clearly allows us to address this challenge.

The results of the ablation are presented in Table 22. Figure 11 illustrates the client count dynamics. We see that for $w > 3$ the communication costs increase significantly. This is also reflected in the client count dynamics: the average value for $m_{\tau+1}$ increases for $w > 3$. We address this to a narrow communication-quality zone from $m = 5$ to $m = 10$ (see the discussion in Appendix D.3). When we choose a resolution of $w = 5$ or more, the ISP technique becomes insensitive to it, which leads to redundant communications at the start, which then generate even more redundant ones due to the greedy structure of the optimized functional equation 3. Thus, the resolution $w$ depends on the statistical indicators of the adaptive intelligence dynamics of the participants. However, for $w < 5$ we do not observe significant deviations, while even $w = 2$ greatly increases the efficiency of internal computations during optimization of equation 3.

We see evidence of the effectiveness of increasing resolution in Table 23. For $w = 3$, the intermediate communication time becomes comparable to the usual one without a significant decrease in performance. At the same time, a subsequent increase in $w$ does not provide such a noticeable speedup in the computations. This is due to the increase in the selected $m_{\tau+1}$ (see Figure 11), which was discussed in the above paragraph.

Table 22: Enumerating resolution $w$ ablation.

| Method | Communications | Test Loss | Accuracy |
|---|---|---|---|
| UNIFORM | 17,767 | 0.461 | 0.842 |
| ISP-UNIFORM $w = 1$ | 14,343 | 0.464 | 0.836 |
| ISP-UNIFORM $w = 2$ | 15,771 | 0.467 | 0.842 |
| ISP-UNIFORM $w = 3$ | 14,515 | 0.483 | 0.837 |
| ISP-UNIFORM $w = 5$ | 19,739 | 0.467 | 0.852 |
| ISP-UNIFORM $w = 7$ | 21,939 | 0.464 | 0.856 |
| ISP-UNIFORM $w = 10$ | 29,984 | 0.458 | 0.860 |

Table 23: Time measurements of ISP intermediate communications in relative to a baseline round across various resolutions $w$.

| Method | Mean Relative Comparison |
|---|---|
| ISP-UNIFORM $w = 1$ | 3.933 |
| ISP-UNIFORM $w = 2$ | 1.814 |
| ISP-UNIFORM $w = 3$ | 1.206 |
| ISP-UNIFORM $w = 5$ | 1.068 |
| ISP-UNIFORM $w = 7$ | 1.011 |
| ISP-UNIFORM $w = 10$ | 1.137 |

Figure 11: Client count dynamics for the ISP technique under various $N$ with UNIFORM sampling strategy. Results are shown up to the best round of each experiment run, as determined by client validation, omitting $\tau + 1/2$ ISP communications.

### E.7  ISP RELATIVE IMPROVEMENT

We revisit the formulation of an intelligent selection of participants equation 3, to emphasize a *relative* communication costs. The communication efficiency nature of optimization equation 3 prioritizes communication reduction by selecting a minimal number $m_{\tau+1}$ of clients that provide positive convergence contributions. Although this approach is effective for overhead minimization, it may sacrifice global convergence optimum. We therefore explore a different adaptive number strategy by explicitly modeling the Quality-Communication trade-off through ISP Relative Improvement (ISP-RI) optimization:

$$m_{\tau+1} = \arg\max_{m} \quad \frac{\mathbb{E}_{C_{\tau+1}} \delta f_\tau(m)}{m^\alpha}, \tag{6}$$

where $\alpha > 0$ is a hyperparameter that determines the relationship in communication-quality trade-off. We set $\alpha = 1$ in the experiment below. This objective function maximizes improvement-per-client rather than merely ensuring positive contributions, naturally shifting focus toward model quality. The implementation of intelligent selection naturally inherits ISP pipeline (Algorithm 2), since the optimization procedure faces the same Problems 1-3 from the original formulation. Note that the iteration loop in Algorithm 2 will be different: optimizing the Relative Improvement requires a complete enumeration of existing $m = \overline{1, M}$, which does not allow for an early exit. Without surrogate loss $\hat{f}$, clients need to participate in these inferences which can lead to their burden. Future

work aims to ablate the complexity reduction techniques, particularly leveraging our method's hyperparameters: Communication frequency $\Delta$ (see Appendix E.2), coverage depth $N$ (see Appendix E.5), and enumerating resolution $w$ (see Appendix E.6).

Table 24: Comparison of ISP-RI with UNIFORM client sampling strategy

| Method | Communications | Test Loss | Accuracy |
|---|---|---|---|
| UNIFORM | 17,767 | 0.461 | 0.842 |
| ADAFL (ECONOM) | 16,191 | 0.508 | 0.827 |
| ADAFL (FULL) | 63,876 | 0.421 | 0.855 |
| ISP UNIFORM | **14,343** | 0.464 | 0.836 |
| ISP-RI UNIFORM | 49,594 | **0.397** | **0.873** |

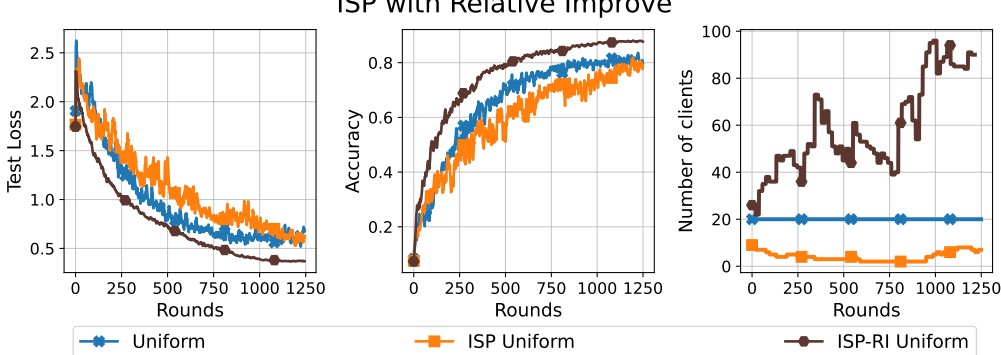

Figure 12: Test loss, accuracy, and client count dynamics, for the ISP and ISP-RI technique with UNIFORM client selection strategy. For clarity, results are shown through round 1250, omitting $\tau + 1/2$ ISP communications.

We compare ISP-RI with baseline UNIFORM client sampling, ISP strategy with functional optimization equation 3, and also with linear increase of ADAFL, since the *full*-schedule (see Appendix D.1.3) addresses the quality trade-off with communication overhead. Table 24 summarizes the results of this comparison. Figure 12) illustrates the convergence dynamic. The ISP-RI approach achieves superiority over all above methods in terms of downstream quality and convergence speed. It also has a positive dynamics of the number of clients as the full-schedule of ADAFL. At the same time, in contrast to the linearity of the baseline schedule, we see a clear nonlinear dependence. As a result, we observe a significant increase in downstream performance with communication savings up to 12%. Furthermore, ISP-RI maintains a more stable convergence. However, ISP-RI is significantly inferior to optimization equation 3 in terms of communications savings, which reflects the communication-quality trade-off that we discussed in Appendix D.3.

Fundamentally, ISP-RI complements the existing results and demonstrates a flexible and extensive mechanism of Our Methodology (see Section 3.2) to the scenarios where attention shifts from communication costs to convergence is critical.

## E.8 RELAXING TRUST ASSUMPTION

Building upon our client computation reduction strategies mentioned in Section 5.2 (specifically partial participation where $m_{\tau+1/2} < M$, analyzed in Appendix E.1), we now examine an orthogonal approach: relaxing requirements for the server-side surrogate loss $\hat{f}$ introduced in Problem 3 and Appendix D.4. While $\hat{f}$ effectively transfers computational burden from clients to the server and accelerates optimization, its assumed availability represents a strong practical constraint. This ablation study quantifies our method's sensitivity to $\hat{f}$ quality, specifically investigating whether performance degradation occurs when using non-ideal surrogate loss functions.

Using the UNIFORM strategy setup from Table 2, we implement two configurations: NORMAL $\hat{f}$: 100-example training subset and CORRUPTED $\hat{f}$: Same subset with aggressive augmentations (for a detailed description of augmentations, see Table 25).

Results (Table 26, Figure 13) demonstrate remarkable consistency: performance metrics and client selection dynamics curves remain nearly identical across standard, corrupted, and $\hat{f}$-free configurations. This minimal sensitivity to $\hat{f}$ quality indicates our method maintains operational integrity even with substantially degraded surrogate $\hat{f}$.

Table 25: Augmentation parameters for corrupted $\hat{f}$

| Augmentation Technique | Parameters | Description |
|---|---|---|
| RANDOMGRAYSCALE | p=0.8 | Randomly converts images to grayscale with high probability. Significantly reduces color information. |
| COLORJITTER | brightness=0.5 contrast=0.5 saturation=0.5 hue=0.3 | Applies strong random color distortions. Severely alters color distribution and visual appearance. |
| RANDOMRESIZEDCROP | size=32 scale=(0.5, 1.0) | Crops random portions of image and resizes. May remove critical features and distort object shapes. |
| RANDOMHORIZONTALFLIP | p=0.5 | Flips image horizontally with 50% probability. Preserves content but alters spatial orientation. |
| RANDOMVERTICALFLIP | p=0.3 | Flips image vertically with 30% probability. Dramatically changes scene orientation (uncommon in natural images). |
| GAUSSIANBLUR | kernel size=3 $\sigma$=(0.1, 2.0) | Adds variable blurring effect. Reduces high-frequency details and sharpness. |
| NORMALIZATION | $\mu$=(0.485, 0.456, 0.406) $\sigma$=(0.229, 0.224, 0.225) | Standard channel-wise normalization. Transforms pixel distributions but preserves visual content. |

Table 26: Surrogate loss $\hat{f}$ ablation.

| Method | Communications | Test Loss | Accuracy |
|---|---|---|---|
| UNIFORM | 17,767 | 0.461 | 0.842 |
| ISP NO $\hat{f}$ | 14,343 | 0.464 | 0.836 |
| ISP NORMAL $\hat{f}$ | 14,525 | 0.462 | 0.837 |
| ISP CORRUPTED $\hat{f}$ | 14,740 | 0.466 | 0.849 |

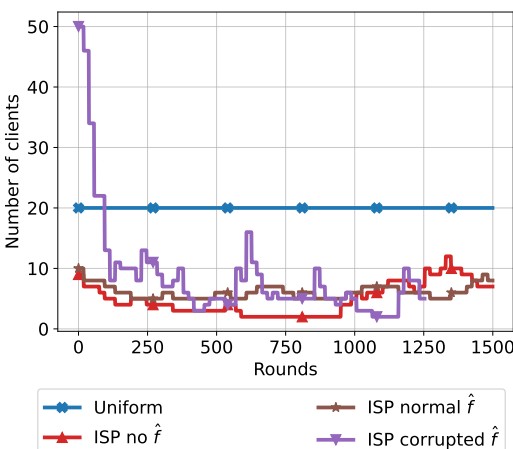

Figure 13: Client count dynamics for the ISP technique with various $\hat{f}$ surrogate loss usage with UNIFORM sampling strategy. For clarity, results are shown through round 1500, omitting $\tau + 1/2$ ISP communications.

## F  THEORETICAL PROOFS

In this section, we provide detailed proofs for the theoretical results presented in the main paper. Specifically, we first prove the Theorem 4.4, which validates our adaptive batch selection strategy, and subsequently derive the global convergence rate (Theorem 4.5).

For the sake of clarity in the theoretical exposition, the analysis below utilizes a unified notation where $m$ denotes the number of participating clients (batch size) and $\mathcal{C}$ denotes the set of sampled clients. This simplification entails no loss of generality. Since our analysis relies on establishing a descent bound on a per-iteration basis, the results naturally extend to the dynamic setting used in Algorithm 2. Specifically, as long as the derived condition on $m$ is satisfied by the chosen $m_{\tau+1}$ at step $\tau$, the descent guarantee holds for that specific transition $\boldsymbol{x}^\tau \to \boldsymbol{x}^{\tau+1}$.

Recall that our analysis operates under Assumptions 4.1 ($L$-smoothness), 4.2 (non-convexity), and 4.3 (bounded gradient heterogeneity with parameter $\rho$).

### F.1  NOTATION

We summarize the key notation used throughout the proofs in the table below:

| Symbol | Description |
|---|---|
| $M$ | Total number of clients available in the network |
| $m$ | Number of clients selected for participation in a given round |
| $B$ | Total local dataset size per client |
| $b$ | Local mini-batch size on each client |
| $\mathfrak{M}$ | Set of indices of the selected clients, $|\mathfrak{M}| = m$ |
| $\mathcal{C}_\tau$ | Set of clients at round $\tau$ (equivalent to $\mathfrak{M}$ in per-round analysis) |
| $f_i(x)$ | Local loss function of client $i$ |
| $f(x)$ | Global objective function, $f(x) = \frac{1}{M} \sum_{i=1}^{M} f_i(x)$ |
| $\nabla f(x)$ | Global gradient at parameters $x$ |
| $\nabla f_i(x, \xi)$ | Stochastic gradient estimate on client $i$ with mini-batch $\xi$ |
| $\bar{g}^\tau$ | Aggregated gradient $\bar{g}^\tau = \frac{1}{m} \sum_{i \in \mathcal{C}_{\tau+1}} \nabla f_i(x^\tau, \xi_i)$ |
| $\sigma^2$ | Bound on the intra-client stochastic gradient variance |
| $\sigma^2_{\text{pop}}$ | Population variance of client updates (inter-client heterogeneity) |
| $\sigma^2_{\text{US}}$ | Sample variance under uniform sampling without replacement |
| $\rho$ | Gradient heterogeneity parameter (Assumption 4.3) |
| $L$ | Smoothness constant of the objective function |
| $\gamma$ | Learning rate (step size) |
| $\alpha$ | Scaling factor for the learning rate, $\gamma \le \alpha/L$ |
| $\delta f_{\tau+1/2}(m)$ | Estimated expected descent with $m$ clients (from Eq. 5) |
| $N$ | Monte Carlo coverage depth (number of samples for estimating $\delta f_\tau$) |
| $\varepsilon$ | $\varepsilon^2$ is an averaged squared gradient norm, $\varepsilon^2 = \frac{1}{T} \sum_{\tau=0}^{T-1} \|\nabla f(x^\tau)\|^2$ |
| $\Delta_T$ | Suboptimality gap, $\Delta_T = f(x^0) - f(x^T)$ |

## F.2 Auxiliary Lemmas

In practical federated learning deployments, the server selects a subset of *distinct* clients for aggregation in each round, which corresponds to uniform sampling *without replacement*. To ensure our theoretical framework rigorously models this setting, we first establish an auxiliary lemmas.

Lemma F.1 allows us to refine the variance term in our convergence analysis.

**Lemma F.1** (Variance for Sampling Without Replacement). *Let $\mathcal{C}$ be a subset of size $m$ sampled uniformly at random without replacement from a population of size $M$. Let $\{X_1, \ldots, X_M\}$ be a set of vectors with mean $\bar{X} = \frac{1}{M} \sum_{i=1}^{M} X_i$. Assume the population variance is bounded by $\sigma^2_{pop}$, i.e., $\frac{1}{M} \sum_{i=1}^{M} \|X_i - \bar{X}\|^2 \le \sigma^2_{pop}$. Then, the variance of the sample mean satisfies:*

$$\mathbb{E}\left[\left\|\frac{1}{m} \sum_{i \in \mathcal{C}} (X_i - \bar{X})\right\|^2\right] \le \sigma^2_{US} \frac{M - m}{m(M - 1)}.$$

*Proof.* For sampling without replacement inside each round, we will use concentration inequalities analogous to those developed by Serfling (1974) and Bardenet & Maillard (2015). Let us define the deviations $d_i := X_i - \bar{X}$ and the sum of squared deviations $A := \sum_{i=1}^{M} \|d_i\|^2$. By the assumption on the population variance, we have:

$$\frac{1}{M} A = \mathbb{E}_{i \sim \text{Unif}[M]} \|X_i - \bar{X}\|^2 \le \sigma^2_{US} \implies A \le M \sigma^2_{US}.$$

Considering $\mathcal{C}$ as a random set of $m$ distinct indices, we define the sample mean deviation $D$:

$$D := \frac{1}{m} \sum_{i \in \mathcal{C}} d_i.$$

We seek to bound $\mathbb{E}\|D\|^2$. Expanding the squared norm yields:

$$\mathbb{E}\|D\|^2 = \frac{1}{m^2} \mathbb{E}\left[\sum_{i,j \in \mathcal{C}} \langle d_i, d_j \rangle\right].$$

The probability that a fixed index $i$ is included in $\mathcal{C}$ is $\frac{m}{M}$. For any pair of distinct indices $i \neq j$, the probability that both are included in $\mathcal{C}$ is $\frac{m(m-1)}{M(M-1)}$. Therefore, we can rewrite the expectation as:

$$\mathbb{E}\|D\|^2 = \frac{1}{m^2}\left(\frac{m}{n}\sum_{i=1}^{M}\|d_i\|^2 + \frac{m(m-1)}{M(M-1)}\sum_{i \neq j}\langle d_i, d_j\rangle\right).$$

Using the identity $\|\sum_{i=1}^{M} d_i\|^2 = \sum_{i=1}^{M}\|d_i\|^2 + \sum_{i \neq j}\langle d_i, d_j\rangle$ and the fact that $\sum_{i=1}^{M} d_i = \sum_{i=1}^{M}(X_i - \bar{X}) = 0$, we find that:

$$\sum_{i \neq j}\langle d_i, d_j\rangle = -\sum_{i=1}^{M}\|d_i\|^2 = -A.$$

Substituting this back into the expression for $\mathbb{E}\|D\|^2$:

$$\mathbb{E}\|D\|^2 = \frac{1}{m^2}\left(\frac{m}{n}A - \frac{m(m-1)}{M(M-1)}A\right)$$
$$= \frac{A}{m^2}\cdot m\left(\frac{1}{M} - \frac{m-1}{M(M-1)}\right)$$
$$= \frac{A}{m}\left(\frac{M-1-(m-1)}{M(M-1)}\right) = \frac{A}{m}\frac{M-m}{M(M-1)}.$$

Finally, applying the bound $A \leq n\sigma_{\mathrm{US}}^2$, we obtain the desired result:

$$\mathbb{E}\left\|\frac{1}{m}\sum_{i \in \mathcal{C}}(X_i - \bar{X})\right\|^2 \leq \frac{M\sigma_{\mathrm{US}}^2}{m}\frac{M-m}{M(M-1)} = \sigma_{\mathrm{US}}^2\frac{M-m}{m(M-1)}.$$

$\square$

According to Lemma F.1,

$$\sigma_{\mathrm{pop}}^2 \leq \sigma_{\mathrm{US}}^2\frac{M-m}{m(M-1)}. \tag{7}$$

### F.3 PROOF OF THEOREM 4.4

We begin by restating the descent guarantee for the ISP procedure. The goal is to show that selecting $m \geq \rho$ devices ensures a strict decrease in the expected objective function value.

**Theorem F.2.** *(**Theorem 4.4**) Under Assumptions 4.1, 4.2, and 4.3, Algorithm 2 with stepsize $\gamma \leq \frac{\alpha}{L}$ (where $\alpha < 2$) and adaptive batch size $m \geq \frac{\alpha\rho}{(2-\alpha)(M-1)+\alpha\rho}M$, guarantees a strictly negative expected descent in the Monte Carlo approximation of the objective function in Equation 5:*

$$\mathbb{E}\delta f_{\tau+1/2}(m_{\tau+1}) = \mathbb{E}\left[\frac{1}{N}\sum_{n=1}^{N}f(x^{(\tau+1/2)n}) - f(x^\tau)\right] < 0.$$

*Proof.* We can proceed with the inequality on the function $f$:

$$f(x^{(\tau+1/2)n}) \leq f(x^\tau) + \langle\nabla f(x^\tau), x^{(\tau+1/2)n} - x^\tau\rangle + \frac{L}{2}\|x^{(\tau+1/2)n} - x^\tau\|^2$$

From the algorithms, we can transform the distance between two points as follows:

$$x^{(\tau+1/2)n} - x^\tau = \frac{\gamma}{m}\sum_{i \in \mathcal{C}_n}(x_i^{\tau+1/2} - x^\tau) = -\frac{\gamma}{m}\sum_{i \in \mathcal{C}_n}\nabla f_i(x^\tau),$$

where $\mathcal{C}_n$ is a group of sampled devices in the $n$-th round of Monte Carlo approximation. Then the inequality transforms into:

$$f(x^{(\tau+1/2)n}) \leq f(x^\tau) - \gamma\langle\nabla f(x^\tau), \frac{1}{m}\sum_{i \in \mathcal{C}_n}\nabla f_i(x^\tau)\rangle + \frac{L\gamma^2}{2}\|\frac{1}{m}\sum_{i \in \mathcal{C}_n}\nabla f_i(x^\tau)\|^2$$

Now we take an expectation with respect to the sampled batch $\mathcal{C}_n$ from both sides of the inequality. Due to random nature of this selection,

$$f(x^\tau) = \mathbb{E}_{\mathcal{C}_n} \frac{1}{m} \sum_{i \in \mathcal{C}_n} f_i(x^\tau) = \mathbb{E} \frac{1}{N} \sum_{n=1}^{N} \frac{1}{m} \sum_{i \in \mathcal{C}_n} f_i(x^\tau).$$

We define

$$\mathbb{E}\hat{f}(x^\tau) := \mathbb{E} \frac{1}{N} \sum_{n=1}^{N} \frac{1}{m} \sum_{i \in \mathcal{C}_n} f_i(x^\tau) \implies \mathbb{E}\hat{f}(x^\tau) = f(x^\tau).$$

Then we get:

$$\mathbb{E}\hat{f}(x^{(\tau+1/2)_n}) \leq f(x^\tau) - \gamma \mathbb{E} \left\langle \nabla f(x^\tau), \nabla \hat{f}(x^\tau) \right\rangle + \frac{L\gamma^2}{2} \mathbb{E}_{\mathcal{C}_n} \left\| \frac{1}{m} \sum_{i \in \mathcal{C}_n} \nabla f_i(x^\tau) \right\|^2.$$

Using law of total expectation, we obtain

$$\begin{aligned}
\mathbb{E}\hat{f}(x^{(\tau+1/2)_n}) &\leq \mathbb{E}f(x^\tau) - \gamma \mathbb{E}\|\nabla f(x^\tau)\|^2 \\
&\quad + \frac{L\gamma^2}{2} \mathbb{E}_{\mathcal{C}_n} \left\| \frac{1}{m} \sum_{i \in \mathcal{C}_n} \nabla f_i(x^\tau) - \nabla f(x^\tau) + \nabla f(x^\tau) \right\|^2 \\
&\leq \mathbb{E}f(x^\tau) - \gamma \mathbb{E}\|\nabla f(x^\tau)\|^2 + \frac{L\gamma^2}{2} \mathbb{E} \|\nabla f(x^\tau)\|^2 \\
&\quad + \frac{L\gamma^2}{2} \frac{M-m}{m(M-1)} \cdot \rho \|\nabla f(x^\tau)\|^2,
\end{aligned}$$

where the last inequality holds according to Lemma F.1, and replacing the standard variance term with $\sigma_{US}^2$ yields a tighter descent condition.

Since $\frac{M-m}{M-1} < 1$, the specific choice of sampling without replacement strictly improves the theoretical convergence properties compared to the baseline assumption. Thus, to have a decrease in a loss function, we need

$$-\gamma \|\nabla f(x^\tau)\|^2 + \frac{L\gamma^2}{2} \|\nabla f(x^\tau)\|^2 + \frac{L\gamma^2}{2} \rho \frac{M-m}{m(M-1)} \|\nabla f(x^\tau)\|^2 < 0.$$

Let us contextualize our convergence target against the baseline of full-batch gradient descent. In the standard non-convex setting, full device participation utilizing a step size $\gamma \leq \frac{\alpha}{L}$ (where $\alpha < 2$). To preserve this asymptotic efficiency despite reduced communication (partial participation), we constrain the step size to be proportional to the inverse smoothness constant:

$$\gamma \leq \frac{\alpha}{L}.$$

Substituting into the desireable descent condition, we have:

$$1 - \frac{L\gamma}{2} \left( 1 + \frac{M-m}{m(M-1)} \cdot \rho \right) \geq 1 - \frac{\alpha}{2} \left( 1 + \frac{M-m}{m(M-1)} \cdot \rho \right) > 0$$

yields the following bounds on the parameters:

$$\alpha < 2 \quad \text{and} \quad m \geq \frac{\alpha\rho}{(2-\alpha)(M-1) + \alpha\rho} \cdot M.$$

This concludes the proof. $\qquad\square$

The inequality $m \geq \frac{\alpha\rho}{(2-\alpha)(M-1)+\alpha\rho} M$ with $\alpha < 2$ derived in the proof reveals a fundamental trade-off between communication efficiency and gradient variance. The parameter $\rho$ acts as a sufficient statistic for the "hardness" of the aggregation step:

- **Adaptive Efficiency:** To guarantee descent, we do not need to aggregate from all $M$ clients. Since in many over-parameterized regimes $\rho \ll M$ (as gradients become aligned), this explains the substantial communication savings observed in practice.

- **Scale Independence:** Crucially, the required batch size $m$ depends on the heterogeneity $\rho$ and its ratio to $M$. This property makes our approach scalable to massive cross-device federated networks where $M \to \infty$ but $\rho$ remains bounded due to the over-parameterization effects discussed in Assumption 4.3.

Having established the descent property per iteration, we now extend this analysis to the global convergence rate. A key question remains: does reducing the communication budget degrade the asymptotic convergence speed compared to full-batch training? The following theorem answers in the negative, showing that our adaptive strategy maintains the standard $\mathcal{O}(1/T)$ rate for non-convex smooth optimization.

### F.4   PROOF OF THEOREM 4.5

**Theorem F.3.** *(**Theorem 4.5**) Under Assumptions 4.1, 4.2, 4.3, after $T$ iterations of Algorithm 2 with $\gamma \leq \frac{\alpha}{L}$ (where $\alpha < 1$) and adaptive number of clients $m \geq \frac{\alpha\rho}{(1-\alpha)(M-1)+\alpha\rho}M$, the averaged squared gradient norm is bounded as:*

$$\frac{1}{T}\sum_{t=0}^{T-1} \mathbb{E}\|\nabla f(x^\tau)\|^2 \leq \mathcal{O}\left(\frac{\Delta_T L}{T}\right).$$

*Proof.* As soon as we choose $m$ devices to communicate from $N$, our gradient estimate is $\frac{1}{m}\sum_{i\in\mathcal{C}} \nabla f_i(x^\tau)$ (Here we denote $\mathcal{C}$ as the batch of the devices chosen). Then we can proceed with the following inequality:

$$f(x^{\tau+1}) \quad \leq \quad f(x^\tau) - \gamma\left\langle \frac{1}{m}\sum_{i\in\mathcal{C}} \nabla f_i(x^\tau), \nabla f(x^\tau)\right\rangle + \frac{L\gamma^2}{2}\left\|\frac{1}{m}\sum_{i\in\mathcal{C}} \nabla f_i(x^\tau)\right\|^2.$$

Now we take an expectation from both sides of the inequality. Due to random nature of this selection, $\mathbb{E}\left[\frac{1}{m}\sum_{i\in\mathcal{C}} \nabla f_i(x^\tau)\right] = \nabla f(x^\tau)$. Then we get:

$$\mathbb{E}f(x^{\tau+1}) \quad \leq \quad f(x^\tau) - \gamma\|\nabla f(x^\tau)\|^2 + \frac{L\gamma^2}{2}\mathbb{E}\left\|\frac{1}{m}\sum_{i\in\mathcal{C}} \nabla f_i(x^\tau)\right\|^2$$

$$= \quad f(x^\tau) - \gamma\|\nabla f(x^\tau)\|^2 + \frac{L\gamma^2}{2}\mathbb{E}\left\|\frac{1}{m}\sum_{i\in\mathcal{C}} (\nabla f_i(x^\tau) - \nabla f(x^\tau) + \nabla f(x^\tau))\right\|^2$$

$$\leq \quad f(x^\tau) - \gamma\|\nabla f(x^\tau)\|^2 + \frac{L\gamma^2}{2}\left\|\frac{1}{m}\sum_{i\in\mathcal{C}} \nabla f(x^\tau)\right\|^2$$

$$+ \frac{L\gamma^2}{2}\mathbb{E}\left\|\frac{1}{m}\sum_{i\in\mathcal{C}} (\nabla f(x^\tau) - \nabla f_i(x^\tau))\right\|^2,$$

where the last inequality holds according to Lemma F.1, replacing the standard variance term with $\sigma_{US}^2$ yields a tighter descent condition.

$$\mathbb{E}f(x^{\tau+1}) \quad \leq \quad f(x^\tau) - \gamma\|\nabla f(x^\tau)\|^2 + \frac{L\gamma^2}{2}\|\nabla f(x^\tau)\|^2 + \frac{L\gamma^2}{2}\frac{M-m}{m(M-1)}\cdot\rho\|\nabla f(x^\tau)\|^2.$$

Before establishing the recurrence relation, we contextualize our convergence target against the baseline of gradient descent. In the standard non-convex setting, full device participation yields a rate of $\mathcal{O}\left(\frac{L\Delta_T}{T}\right)$ utilizing a step size $\gamma \leq \frac{\alpha}{L}$ (where $\alpha \leq 1$). To preserve this asymptotic efficiency despite reduced communication (partial participation), we constrain the step size to be proportional to the inverse smoothness constant:

$$\gamma \leq \frac{\alpha}{L}.$$

Substituting this parameterization into the descent inequality allows us to proceed with the recursion:

$$-\gamma \|\nabla f(x^\tau)\|^2 + \frac{L\gamma^2}{2}\|\nabla f(x^\tau)\|^2 + \frac{L\gamma^2}{2}\rho\frac{M-m}{m(M-1)}\|\nabla f(x^\tau)\|^2 \leq -\frac{1}{2}.$$

Then the desireable descent condition is the following:

$$1 - \frac{L\gamma}{2}\left(1 + \rho\frac{M-m}{m(M-1)}\right) \geq 1 - \frac{\alpha}{2}\left(1 + \rho\frac{M-m}{m(M-1)}\right) \geq \frac{1}{2}.$$

We set

$$\alpha < 1 \quad \text{and} \quad m \geq \frac{\alpha\rho}{(1-\alpha)(M-1) + \alpha\rho} \cdot M.$$

Now we have:

$$f(x^{\tau+1}) \leq f(x^\tau) - \frac{\gamma}{2}\|\nabla f(x^\tau)\|^2.$$

Therefore:

$$f(x^T) \leq f(x^0) - \frac{\gamma}{2}\sum_{\tau=0}^{T-1}\|\nabla f(x^\tau)\|^2.$$

Rearranging and defining $\varepsilon^2 = \frac{1}{T}\sum_{\tau=0}^{T-1}\|\nabla f(x^\tau)\|^2$:

$$\varepsilon^2 \leq \frac{2(f(x^0) - f(x^T))}{\gamma T} < \frac{4L\Delta_T}{T},$$

where $\Delta_T = f(x^0) - f(x^T)$ is the suboptimality gap.

This establishes the convergence rate:

$$\boxed{\varepsilon^2 \leq \mathcal{O}\left(\frac{L\Delta_T}{T}\right),}$$

which matches the rate for smooth non-convex optimization, while communicating with only $m = \mathcal{O}(\rho)$ devices per round instead of all $M$ devices. When $\rho \ll M$, this represents the $\mathcal{O}(M/\rho)$-fold reduction in communication cost per iteration. $\qquad\square$

**Discussion.** Our analysis incorporates the exact finite population correction for sampling without replacement, resulting in a precise lower bound for $m$. While the condition appears complex, it reveals insights into the interplay between communication, heterogeneity, and convergence speed.

**1. Asymptotic Scalability ($M \to \infty$).** Consider the regime of large-scale federated networks where the total number of clients $M$ is large. If we fix the step-size parameter $\alpha < 1$, the term $(1-\alpha)(M-1)$ dominates the denominator. Consequently, the bound simplifies asymptotically to:

$$m \gtrsim \frac{\alpha}{1-\alpha}\rho.$$

This confirms that the required active cohort size $m$ scales linearly with data heterogeneity $\rho$ and remains **independent of the total population size** $M$ (as long as $M \gg \rho$). This ensures that our method scales efficiently to massive cross-device setups without requiring a proportional increase in communication.

**2. The Step-Size vs. Communication Trade-off.** The parameter $\alpha$ directly controls the learning rate $\gamma$. The derived bound illustrates a flexible design space:

- *Faster Convergence:* As $\alpha \to 1$, the denominator shrinks, requiring $m \to M$. This implies that to use the maximum possible step size (most aggressive convergence), one must approach full participation.
- *Communication Efficiency:* Conversely, choosing a smaller $\alpha$ allows for a significantly smaller $m$. Our theory guarantees that even with a reduced step size, the algorithm maintains the $\mathcal{O}(1/T)$ rate, but with a much smaller communication footprint per round.

**3. Efficiency in Over-parameterized Regimes.** Finally, in the over-parameterized settings discussed in Assumption 4.3, local gradients tend to align, implying $\rho \ll M$. Under this condition, the fraction of required devices $\frac{m}{M}$ becomes negligible. Thus, ISP achieves the classical non-convex convergence rate while reducing communication costs by a factor of roughly $\mathcal{O}(M/\rho)$, providing a rigorous justification for the empirical savings.

### F.5 STOCHASTIC SETTING

In practical Federated Learning scenarios, local datasets are often too large to process in full batches, necessitating the stochasticity. While the deterministic analysis provides a foundational understanding of ISP's mechanism, it does not account for the noise introduced by mini-batch sampling on client devices. This noise, combined with client drift, complicates the convergence landscape. Therefore, we extend our theoretical framework to the stochastic setting to demonstrate that our adaptive client selection strategy remains robust even when clients return noisy gradient estimates.

We formalize the stochastic nature of local updates with the following standard assumption, which bounds the variance of the gradient estimation.

**Assumption F.4.** Each worker has access to an independent and unbiased stochastic gradient $\nabla f_{\xi_i}(x) = \nabla f_i(x, \xi_i) = \nabla f_i(x, \xi_i)$ with $\mathbb{E}[\nabla f_i(x, \xi_i)] = \nabla f_i(x)$ and its variance is bounded by $\sigma^2$:

$$\mathbb{E}\|\nabla f_i(x, \xi_i) - \nabla f_i(x)\|^2 \leqslant \sigma^2, \quad \text{for all } x \in \mathbb{R}^d.$$

With this assumption, we derive the convergence guarantee for the non-convex stochastic regime. The following theorem establishes that ISP maintains the standard $\mathcal{O}(1/\sqrt{T})$ rate while significantly reducing the communication overhead.

**Theorem F.5.** *Under Assumptions 4.1, 4.2, 4.3, F.4 after $T$ iterations of Algorithm 2 with $\gamma = \min\{\frac{\alpha}{L}, \sqrt{\frac{\Delta_T}{L\sigma^2 T}}\}$ (where $\alpha < \frac{1}{2}$) and adaptive number of clients $m \geq \frac{2\alpha\rho}{(1-2\alpha)(M-1)+2\alpha\rho}M$, the averaged squared gradient norm is bounded as:*

$$\frac{1}{T}\sum_{t=0}^{T-1}\mathbb{E}\|\nabla f(x^\tau)\|^2 \leq \mathcal{O}\left(\frac{L\Delta_T}{T} + \sigma\sqrt{\frac{L\Delta_T}{T}}\right).$$

*Proof.* We begin with the $L$-smoothness property of the objective function. For a generic iteration $\tau$, the update rule implies:

$$f(x^{\tau+1}) \leq f(x^\tau) + \langle \nabla f(x^\tau), x^{\tau+1} - x^\tau \rangle + \frac{L}{2}\|x^{\tau+1} - x^\tau\|^2$$

$$= f(x^\tau) - \gamma\langle \nabla f(x^\tau), \bar{g}^\tau \rangle + \frac{L\gamma^2}{2}\|\bar{g}^\tau\|^2,$$

where $\bar{g}^\tau = \frac{1}{m}\sum_{i\in\mathcal{C}_{\tau+1}}\nabla f_i(x^\tau, \xi_i)$ is the aggregated stochastic gradient. Taking the expectation over the random sampling of clients $\mathcal{C}_{\tau+1}$ and local stochastic samples $\xi$, and noting that $\mathbb{E}[\bar{g}^\tau] = \nabla f(x^\tau)$, we obtain:

$$\mathbb{E}f(x^{\tau+1}) \leq f(x^\tau) - \gamma\|\nabla f(x^\tau)\|^2 + \frac{L\gamma^2}{2}\mathbb{E}\left\|\frac{1}{m}\sum_{i\in\mathcal{C}_{\tau+1}}\nabla f_i(x^\tau, \xi_i)\right\|^2.$$

To bound the last term, we decompose the variance into components arising from client sampling and stochastic noise. Using the inequality $\|a + b + c\|^2 \leq 3(\|a\|^2 + \|b\|^2 + \|c\|^2)$:

$$\mathbb{E}\|\bar{g}^\tau\|^2 = \mathbb{E}\left\|\frac{1}{m}\sum_{i\in\mathcal{C}_{\tau+1}}((\nabla f_i(x^\tau, \xi_i) - \nabla f_i(x^\tau)) + (\nabla f_i(x^\tau) - \nabla f(x^\tau)) + \nabla f(x^\tau))\right\|^2$$

$$\leq 2\mathbb{E}\left\|\frac{1}{m}\sum_{i\in\mathcal{C}_{\tau+1}}(\nabla f_i(x^\tau, \xi_i) - \nabla f_i(x^\tau))\right\|^2$$

$$+2\mathbb{E}\left\|\frac{1}{m}\sum_{i\in\mathcal{C}_{\tau+1}}\left(\nabla f_i(x^\tau)-\nabla f(x^\tau)\right)+\nabla f(x^\tau)\right\|^2$$

$$\leq\quad 2\mathbb{E}\left\|\frac{1}{m}\sum_{i\in\mathcal{C}_{\tau+1}}\left(\nabla f_i(x^\tau,\xi_i)-\nabla f_i(x^\tau)\right)\right\|^2$$

$$+2\mathbb{E}\left\|\frac{1}{m}\sum_{i\in\mathcal{C}_{\tau+1}}\left(\nabla f_i(x^\tau)-\nabla f(x^\tau)\right)\right\|^2+2\|\nabla f(x^\tau)\|^2.$$

The first term relates to the stochastic noise. Due to the independence of mini-batches, it is bounded by $\frac{\sigma^2}{m}\leq\sigma^2$. The second term captures the variance due to client sampling. Applying Lemma F.1 (variance of sampling without replacement) and Assumption 4.3 ($\rho$-heterogeneity), we have:

$$\mathbb{E}\left\|\frac{1}{m}\sum_{i\in\mathcal{C}_{\tau+1}}\left(\nabla f_i(x^\tau)-\nabla f(x^\tau)\right)\right\|^2=\frac{M-m}{m(M-1)}\sigma_{pop}^2\leq\frac{M-m}{m(M-1)}\rho\|\nabla f(x^\tau)\|^2.$$

Substituting these bounds back into the descent inequality yields:

$$\mathbb{E}f(x^{\tau+1})\leq f(x^\tau)-\gamma\|\nabla f(x^\tau)\|^2+L\gamma^2\left[\|\nabla f(x^\tau)\|^2+\frac{M-m}{m(M-1)}\rho\|\nabla f(x^\tau)\|^2+\sigma^2\right].$$

To ensure a monotonic decrease in the objective function (ignoring the irreducible $\sigma^2$ term), we require the coefficient of $\|\nabla f(x^\tau)\|^2$ to be sufficiently negative, specifically less than $-\frac{\gamma}{2}$. This leads to the condition:

$$-\gamma+L\gamma^2\left(1+\rho\frac{M-m}{m(M-1)}\right)\leq-\frac{\gamma}{2}\iff 1-2L\gamma\left(1+\rho\frac{M-m}{m(M-1)}\right)\geq 0.$$

Letting $\gamma=\frac{\alpha}{L}$, this inequality holds if $1-2\alpha(1+\rho\frac{M-m}{m(M-1)})\geq 0$. Solving for $m$ yields the required adaptive batch size:

$$m\geq\frac{2\alpha\rho}{(1-2\alpha)(M-1)+2\alpha\rho}M.$$

Setting $\alpha<1/2$ ensures the denominator is positive. Under this condition, the recursion simplifies to:

$$\mathbb{E}f(x^{\tau+1})\leq f(x^\tau)-\frac{\gamma}{2}\|\nabla f(x^\tau)\|^2+L\gamma^2\sigma^2.$$

Summing over $\tau=0,\ldots,T-1$ and rearranging gives:

$$\frac{1}{T}\sum_{\tau=0}^{T-1}\mathbb{E}\|\nabla f(x^\tau)\|^2\leq\frac{2(f(x^0)-f(x^T))}{\gamma T}+2L\gamma\sigma^2.$$

To balance the deterministic and stochastic error terms, we choose $\gamma\sim\frac{1}{\sqrt{T}}$. Specifically, setting $\gamma=\min\{\frac{\alpha}{L},\sqrt{\frac{\Delta_T}{L\sigma^2T}}\}$ yields the final rate:

$$\boxed{\frac{1}{T}\sum_{\tau=0}^{T-1}\mathbb{E}\|\nabla f(x^\tau)\|^2\leq\mathcal{O}\left(\frac{L\Delta_T}{T}+\frac{\sigma\sqrt{L\Delta_T}}{\sqrt{T}}\right).}$$

This result matches the optimal convergence rate for non-convex stochastic optimization. Crucially, this rate is achieved while communicating with only $m\approx\mathcal{O}(\rho)$ clients per round in the low-heterogeneity regime ($\rho\ll M$). This confirms that ISP effectively trades off redundant communication for computation without compromising the asymptotic convergence speed. □

MINI-BATCH EFFECT

While the stochastic analysis in Section F.5 assumes access to an infinite data stream (standard SGD setting), real-world Federated Learning clients typically hold a fixed, finite dataset of size $B$. In this regime, the local update is computed by sampling a mini-batch of size $b < B$ *without replacement*. This distinction is crucial: as $b$ approaches $B$, the variance of the gradient estimate should vanish, recovering the deterministic full-batch gradient descent behavior. The standard stochastic assumption fails to capture this interpolation.

To address this, we refine our analysis by modeling the local gradient as a sample from a finite population. This introduces a finite population correction factor that reduces the variance bound as the mini-batch size increases.

**Theorem F.6.** *Under Assumptions 4.1, 4.2, 4.3, F.4 and assuming each client holds a finite dataset of size $B$ with bounded gradient variance $\sigma^2$, let Algorithm 2 run for $T$ iterations. If we employ a step size $\gamma = \min\{\frac{\alpha}{L}, \sqrt{\frac{\Delta_T}{L\sigma^2 T} \cdot \frac{b(B-1)}{B-b}}\}$ (where $\alpha < \frac{1}{2}$ and $b < B$), and the adaptive number of clients satisfies the heterogeneity-aware condition $m \geq \frac{2\alpha\rho}{(1-2\alpha)(M-1)+2\alpha\rho}M$, then the averaged squared gradient norm is bounded as:*

$$\frac{1}{T}\sum_{\tau=0}^{T-1}\mathbb{E}\|\nabla f(x^\tau)\|^2 \leq \mathcal{O}\left(\frac{L\Delta_T}{T} + \sigma\sqrt{\frac{L\Delta_T}{T} \cdot \underbrace{\frac{B-b}{b(B-1)}}_{\textit{Finite Correction}}}\right).$$

*Proof.* We proceed similarly to the proof of Theorem F.5, starting from the $L$-smoothness expansion and taking the total expectation. The critical difference lies in the decomposition of the variance term $\mathbb{E}\|\bar{g}^\tau - \nabla f(x^\tau)\|^2$. Recall that the aggregated gradient is $\bar{g}^\tau = \frac{1}{m}\sum_{i\in\mathcal{C}_{\tau+1}}\nabla f_i(x^\tau, \xi_i)$. We decompose the error into inter-client variance (client sampling) and intra-client variance (data sampling):

$$
\begin{aligned}
\mathbb{E}\|\bar{g}^\tau - \nabla f(x^\tau)\|^2 &= \mathbb{E}\left\|\frac{1}{m}\sum_{i\in\mathcal{C}_{\tau+1}}(\nabla f_i(x^\tau) - \nabla f(x^\tau))\right\|^2 \\
&\quad + \mathbb{E}\left\|\frac{1}{m}\sum_{i\in\mathcal{C}_{\tau+1}}(\nabla f_i(x^\tau, \xi_i) - \nabla f_i(x^\tau))\right\|^2.
\end{aligned}
$$

1. **Inter-client Variance:** As established in Theorem F.5, applying Lemma F.1 (variance of sampling $m$ clients from $M$ without replacement) combined with $\rho$-heterogeneity yields:

$$\mathbb{E}\left\|\frac{1}{m}\sum_{i\in\mathcal{C}_{\tau+1}}(\nabla f_i(x^\tau) - \nabla f(x^\tau))\right\|^2 \leq \frac{M-m}{m(M-1)}\rho\|\nabla f(x^\tau)\|^2.$$

2. **Intra-client Variance:** For the second term, we consider the randomness of the local mini-batch $\xi_i$ of size $b$ sampled from the local dataset of size $B$ without replacement. Using the property of independence between clients and applying the finite population correction (Lemma F.1) locally:

$$\mathbb{E}\|\nabla f_i(x^\tau, \xi_i) - \nabla f_i(x^\tau)\|^2 \leq \sigma^2\frac{B-b}{b(B-1)}.$$

Averaging this over $m$ clients (and noting that the variance scales by $1/m$ for independent zero-mean variables) gives:

$$\mathbb{E}\left\|\frac{1}{m}\sum_{i\in\mathcal{C}_{\tau+1}}(\nabla f_i(x^\tau, \xi_i) - \nabla f_i(x^\tau))\right\|^2 \leq \sigma^2\frac{B-b}{b(B-1)}.$$

Note: We conservatively dropped the $1/m$ factor in the final inequality to unify the bound, though keeping it would only tighten the result.

Substituting these two variance components back into the descent inequality:

$$\mathbb{E}f(x^{\tau+1}) \leq f(x^\tau) - \gamma\|\nabla f(x^\tau)\|^2$$
$$+ L\gamma^2\left[\|\nabla f(x^\tau)\|^2 + \frac{M-m}{m(M-1)}\rho\|\nabla f(x^\tau)\|^2 + \frac{B-b}{b(B-1)}\sigma^2\right].$$

To guarantee monotonic descent (modulo the noise term), we enforce the same condition on $m$ as in Theorem F.5, ensuring the gradient coefficients sum to $\leq -\gamma/2$. This requires $\gamma \leq \alpha/L$ and $m \geq \frac{\alpha\rho}{(1/2-\alpha)(M-1)+\alpha\rho}M$. This simplifies the recursion to:

$$\mathbb{E}f(x^{\tau+1}) \leq \mathbb{E}f(x^\tau) - \frac{\gamma}{2}\mathbb{E}\|\nabla f(x^\tau)\|^2 + L\gamma^2\sigma^2\frac{B-b}{b(B-1)}.$$

Summing over $T$ iterations and rearranging for the average gradient norm $\varepsilon^2 = \frac{1}{T}\sum_{\tau=0}^{T-1}\mathbb{E}\|\nabla f(x^\tau)\|^2$:

$$\varepsilon^2 \leq \frac{2(f(x^0)-f(x^T))}{\gamma T} + 2L\gamma\sigma^2\frac{B-b}{b(B-1)}.$$

We optimize the step size $\gamma$ to balance the deterministic and stochastic error terms. Setting the two terms on the RHS to be equal yields the optimal step size:

$$\gamma = \frac{1}{\sigma}\sqrt{\frac{\Delta_T}{LT}\cdot\frac{b(B-1)}{B-b}}.$$

Substituting this $\gamma$ back into the bound yields the final result:

$$\boxed{\varepsilon^2 \leq \mathcal{O}\left(\frac{L\Delta_T}{T} + \sigma\sqrt{\frac{L\Delta_T}{T}\frac{B-b}{b(B-1)}}\right).}$$

$\square$

**Discussion.** This theorem provides a precise characterization of how local computation affects global convergence. The term $\frac{B-b}{b(B-1)}$ acts as a "variance reduction switch."

- In the **Stochastic limit** ($b \ll B$), the term approaches $1/b$, recovering the standard stochastic convergence rate dominated by sampling noise.
- In the **Full-batch limit** ($b \to B$), the term vanishes. This implies that if clients perform full-batch updates, the stochastic noise is eliminated, and the algorithm recovers the fast convergence rates of deterministic gradient descent, limited only by the client heterogeneity handled by our adaptive $m$. We note that if $b = B$, we can apply the standard analysis and obtain an estimate without such variance at all.

This result theoretically justifies using larger local batches to offset the noise introduced by aggressive communication compression (small $m$).

THE USE OF LARGE LANGUAGE MODELS (LLMs)

In this work, Large Language Models (LLMs) were used exclusively for spelling edits.

