# OpenReview forum: "Communication-Efficient Federated Learning with Adaptive Number of Participants"
_ICLR.cc/2026/Conference — Submitted to ICLR 2026_

### Official Review · Reviewer_hF4z · 2025-10-26

**Soundness:** 3
**Presentation:** 3
**Contribution:** 2
**Rating:** 4
**Confidence:** 4

**Summary:**

The paper introduces Intelligent Selection of Participants (ISP), a dynamic mechanism that adaptively determines the optimal number of clients to involve in each communication round of federated learning. Unlike conventional approaches such as FedAvg and FedProx that assume a fixed client count, ISP formulates participant selection as an optimization problem to minimize communication costs while maintaining model improvement. It integrates seamlessly with standard federated algorithms and operates without modifying client optimizers. Extensive experiments on CIFAR-10, Tiny-ImageNet, and real-world ECG data show that ISP consistently reduces communication costs by up to 30%, without sacrificing model accuracy.

**Strengths:**

+ Introduces the Intelligent Selection of Participants (ISP) framework, which formulates client-count determination as an optimization problem balancing convergence and communication cost.
+ Empirically validated across diverse datasets (CIFAR-10, Tiny-ImageNet, and ECG), showing up to 30% communication savings and no degradation in accuracy.
+ Offers a general solution applicable to both standard and advanced FL scenarios, including gradient compression.
+ The paper is well-structured and easy to follow.

**Weaknesses:**

- The paper’s major weakness lies in its lack of theoretical analysis. Despite formulating client-count selection as an optimization problem, the authors provide no convergence guarantees, communication–computation trade-off bounds, or proofs of optimality for the ISP mechanism.
- The optimization in Equation (3) is treated heuristically via Monte Carlo approximation, without analytical discussion of its stability, variance, or expected bias.
- Without formal complexity or asymptotic analysis, it remains unclear whether the observed 30% communication savings are due to principled algorithmic efficiency or empirical tuning effects.
- The intermediate communication step (τ + ½) introduces synchronization and computation overhead, which might partially offset communication savings, especially in large or unstable networks. However, this overhead is only discussed qualitatively and not measured quantitatively.
- The Monte Carlo estimation of loss reduction has no analysis of its variance, bias, or sample efficiency, raising concerns about ISP’s decision reliability across rounds.
- In the CIFAR-10 experiments, results show that POW-D outperforms ISP-POW-D in terms of test loss and communication cost balance. This inconsistency undermines claims that ISP universally enhances all sampling strategies.
- The paper does not include statistical significance testing of improvements, leaving uncertainty over whether the reported gains (especially marginal ones) are meaningful.
- The evaluation scope is narrow, focusing on image and ECG classification tasks, and other domains like NLP are not explored, limiting the method’s generalizability.
- Experimental settings (e.g., client sampling distributions, training parameters) are split between the main text and appendices, reducing accessibility and reproducibility for readers.

**Questions:**

1. Can you provide any theoretical justification or convergence analysis for ISP? Even partial proofs or convergence bounds would strengthen the paper.
2. How stable is the Monte Carlo loss estimation? Include results showing its variance or reliability across runs.
3. How costly is the intermediate step (τ + ½) in practice? Please measure and report this overhead quantitatively.
4. Are the 30% communication savings due to ISP’s design or specific hyperparameter tuning?
5. Why does POW-D outperform ISP-POW-D on CIFAR-10? Analyze when ISP may underperform or over-select clients.
6. Can you test ISP on other domains (e.g., NLP or speech tasks)? This would show its generality beyond image and ECG data.
7. How would ISP perform under client dropouts or asynchronous updates? A short sensitivity test would make the results more practical.
8. Discuss how ISP could work with differential privacy or personalized FL frameworks.

---

> ### Author Response · Authors · 2025-11-21
>
> We thank Reviewer hF4z for their detailed feedback. We hope our clarifications will help fully resolve any outstanding questions regarding our work. To ensure clarity, we have grouped the raised weaknesses and questions by topic and address them collectively below. Throughout our response, we refer to the revised version of the paper, which we have uploaded to OpenReview. For convenience, all major additions and revisions in the manuscript are highlighted in blue.
>
> > W1/Q1: Theoretical analysis
>
> We sincerely appreciate the reviewer’s emphasis on theoretical rigor. We dedicated the revision period to formalizing the mathematical foundation of ISP.
>
> To this end, we have introduced Theorem 4.4, which derives an explicit lower bound for the participation size $m_{\tau+1}$. This bound is the precise condition required to guarantee a strict reduction in the global loss function at each step. Furthermore, Theorem 4.5 proves that ISP maintains the standard non-convex convergence rate of $\mathcal{O}(1/T)$. This confirms that our dynamic reduction of clients does not come at the cost of stability or asymptotic convergence speed compared to fixed-size baselines.
>
> Perhaps most importantly, our analysis uncovers the physical intuition driving the method: the required participation $m$ scales with data heterogeneity ($\rho$) rather than the total network size ($M$). In simple terms, we provide a formal justification that in regimes with moderate heterogeneity (low $\rho$), the theoretical need for massive participation disappears, simplifying to $m = \mathcal{O}(\rho)$. This insight fundamentally shifts the selection of $m$ from a heuristic guess to a principled decision governed by the complexity of the loss landscape.
>
> > W2/Q2/W5: Monte-Carlo estimation
>
> The Monte Carlo loss estimation used in ISP is stable and well-controlled by the hyperparameter $N$. As demonstrated in our ablation study (Appendix E.5, Table 16), even relatively small values of $N$ produce consistent loss estimates across runs, with only minor fluctuations in both communication savings and model performance.
>
> Crucially, the Monte Carlo estimation can be performed primarily on the server side, eliminating additional computational overhead on edge clients while ensuring stable and reliable improvement measurements. This design choice balances both practical applicability and theoretical robustness.
>
> > W3/Q4: 30% communication savings
>
> The observed 30%+ communication savings stem primarily from the principled design of ISP. As demonstrated in our ablation study (Appendix E), tuning the hyperparameter $m_{\tau + 1/2}$ can bring slight additional improvements beyond the results reported in Table 2, but the overall communication savings and accuracy trends remain consistent across various hyperparameter settings.
>
> All core ISP parameters—$\omega$, $\Delta$, and $N$—are simple, interpretable, and robust across a wide parameter range, mainly controlling the granularity and depth of improvement estimation. Default values used in the main experiments reside well within stable regions identified by ablation, ensuring the communication gains are systematic and reproducible, not dependent on hyperparameter overfitting. This confirms that the communication efficiency is inherent to ISP's algorithmic framework rather than resulting from empirical tuning.
>
> > W4/Q3 Overhead of intermediate step $\tau + \frac{1}{2}$
>
> While Algorithm 2 presents the intermediate step as a full-client round for conceptual clarity, ISP does not obligatorily require full participation in practice. Table 12 demonstrates that the method remains stable–and even reduces total communication–when significantly fewer clients participate in the intermediate round. This directly models real large-scale FL environments where clients routinely fail to respond due to connectivity issues or resource constraints.
>
> Quantitatively, the intermediate step accounts for about 30% of total communication volume on average. However, the intermediate step is invoked only once every $\Delta$ rounds, and $\Delta$ can be set large to further amortize its cost with no loss in stability (see Appendix E). Our ECG experiment with 2,000 clients (Table 4) reinforces this: despite large-scale participation, ISP delivers significant communication savings and only benefits from scalability. Thus, although the intermediate step adds overhead, it is controlled and outweighed by total communication savings even in large-scale deployments.
>
> Please, see our next comment where we address other Weaknesses / Questions.

---

> ### Author Response · Authors · 2025-11-21
>
> > W6/Q5: Pow-D performance
>
> We explicitly described this observation in Section 5.2. As stated in the paper: “Only for the \textsc{POW-D} baseline, the \textsc{ISP} technique requires barely more communication but demonstrates a substantial increase in the target metrics that we attribute to the adversarial nature of client sampling in \textsc{POW-D}.”
> Indeed, POW-D client selection algorithm tends to prioritize clients with the highest validation loss, which, under high heterogeneity, may result in the selection of clients that degrade model performance -- close to Byzantine behavior in FL. To mitigate this issue, our proposed ISP method requires a larger number of participating clients, thereby improving the robustness and final performance of the trained model.
> It is also important to note that if POW-D were configured with a larger fixed number of clients per round, it could achieve final performance comparable to ISP–POW-D. However, this would come at the cost of substantially higher communication, since increasing the round-wise participant count directly increases communication overhead. We intentionally fixed the number of clients to m = 20 for all client-sampling baselines, following the careful analysis presented in Section D.2 (Table 7), in order to provide a fair and consistent comparison.
> The POW-D case represents a trade-off where the modest communication increase is offset by meaningful performance gains, confirming ISP’s overall efficiency advantage.
> > W7: Statistical significance
> We report the results averaged over five independent runs in Table 2, and we provide the corresponding variability (±) for communication, accuracy, and test loss in each entry. We believe this allows readers to assess the consistency of our findings and provides a practical sense of the reliability of the reported improvements, including the marginal ones.
>
> > W8/Q6: Narrow evaluation
>
> We respectfully disagree that the evaluation scope is narrow. Our experimental setup is comprehensive and systematically validates ISP across diverse conditions. Specifically, we evaluate on multiple datasets spanning distinct domains: CIFAR-10 and Tiny-ImageNet for vision tasks, plus the large-scale PTB-XL ECG dataset for medical time-series classification. The ECG experiments involve 2,000 clients, demonstrating scalability far beyond typical FL benchmarks. We conduct extensive ablation studies (Appendix E) covering all critical hyperparameters–$\omega$, $\Delta$, $N$, and $m_{\tau+1/2}$--as well as intermediate round participation (Table 12) and Monte Carlo stability (Table 16). Our evaluation includes comparisons with multiple state-of-the-art baselines (Uniform, POW-D, FedCor, FedCBS, DELTA) under varied data heterogeneity settings, vision transformers (Swin-T), and gradient compression methods (QSGD with TopK and RandK). All results are averaged over five independent runs with reported variance, ensuring statistical rigor. We also provide formal convergence analysis (Theorem F.4) establishing theoretical guarantees.
>
> Due to the limited rebuttal period, we cannot complete full-scale NLP experiments within the available timeframe. However, ISP operates on fundamental FL primitives–gradient updates, loss estimation, and client selection–that are inherently domain-agnostic. The method does not exploit any vision-specific or ECG-specific properties, making the demonstrated communication savings and convergence behavior transferable across modalities. We believe the current evaluation comprehensively validates ISP's efficacy and provides a complete, rigorous experimental pipeline for the FL community.
>
> > W9:Description of experimental setting
>
> All the hyperparameters can be found in one place - Table 6 in the Appendix to avoid overloading the main text. We also introduced them in the main text where they appeared for the first time. However, for the convenience of readers, we included a reference, thanks for the suggestion.
> > Q7:
> ISP is designed for synchronous FL and does not address asynchronous aggregation–a substantial research area with many dedicated works. Our contribution focuses on a distinct dimension: dynamically selecting how many clients to involve, orthogonal to communication protocol design.
>
> That said, ISP's core mechanism could be adapted to asynchronous settings. The improvement estimation could be reformulated to operate over clients whose updates arrive within a time window, enabling integration with asynchronous pipelines. In our ablation study (Table 12), the method maintains stable performance even when only a fraction of clients contribute updates in the $\tau + 1/2$ round; this can be interpreted as a proxy for common system heterogeneity, such as client unavailability or network issues. However, we highlight that this extension is outside the current scope.
>
> Please, see our next comment where we address the last Question.

---

> ### Author Response · Authors · 2025-11-21
>
> >Q8: Сompatibility with differential privacy and personalized FL frameworks
>
> Differential privacy represents a distinct research area with its own implementation challenges. Standard federated algorithms–including FedAvg, FedProx, SCAFFOLD, and FedNova–transmit gradient updates or model parameters, which are not inherently private without additional cryptographic or noise-injection mechanisms. ISP operates orthogonally to privacy-preserving techniques and does not aim to provide a unified framework addressing all federated learning challenges simultaneously. Integrating differential privacy would require additional noise calibration mechanisms that extend beyond our current scope.
>
> Personalized federated learning, however, presents a more natural integration path. Methods such as FedPer, Ditto, pFedMe, and APFL [1,2,3,4] cluster clients with similar data distributions or train client-specific model components. While full exploration of this direction lies outside our current scope, we note that ISP's adaptive selection mechanism could operate effectively within personalized clusters. Specifically, when personalized FL algorithms partition clients into cohorts, ISP could be applied independently to each cluster, dynamically determining the optimal number of participants per cohort. This would preserve ISP's communication efficiency while respecting personalized learning objectives.
>
> ---
> Overall, we sincerely appreciate the detailed discussion raised by Reviewer hF4z. Following these clarifications, we believe our contribution is now complete and well-understood. We have added formal convergence analysis that significantly strengthens the theoretical foundation (see Section 4 and Appendix F), while our experimental section comprehensively validates ISP across diverse datasets, 2,000 clients, extensive ablations, multiple baselines, and statistical rigor. While additional domains (e.g., NLP) are constrained by the rebuttal timeline, we believe the current evaluation systematically demonstrates ISP's efficacy on domain-agnostic FL primitives. Regarding future directions such as asynchronous aggregation and personalized FL - we find these questions interesting and have outlined potential integration paths, though they lie outside the current scope. We respectfully ask the reviewer to reconsider the evaluation in light of these substantial additions and clarifications.
>
> [1] Arivazhagan M. G. et al. Federated learning with personalization layers arXiv preprint arXiv:1912.00818. – 2019.
> [2] Li T. et al. Ditto: Fair and robust federated learning through personalization ICML 2021
> [3] T Dinh C., Tran N., Nguyen J. Personalized federated learning with moreau envelopes NeurIPS 2020
> [4] Fan K. et al. APFL: Analytic Personalized Federated Learning via Dual-Stream Least Squares arXiv 2025.

---

> > ### Comment · Reviewer_hF4z · 2025-11-22
> >
> > I thank the authors for the theoretical analysis and for addressing my previous comments. I will summarize all of my comments below.
> >
> > For the theoretical analysis, I have a few questions:
> > - How is the Monte-Carlo estimation considered in the proof in Theorem  4.4?  In the proof of Theorem 4.5, it considers full device participation under full-batch gradient descent and uses Theorem 4.4 to prove the convergence.
> > - Based on the theoretical analysis, the selection of m requires the knowledge of heterogeneity ρ. How can we obtain ρ in practice? In the experiments, it does not discuss how m is selected. Moreover, the theoretical results require m larger than a threshold, but the experiments show that the choice of m is small in certain rounds (e.g., Fig. 2, Fig. 5).
> > - ρ is defined as gradient heterogeneity. The authors claim that it is data heterogeneity in the response. It should be noted that the effect of mini-batch size on gradient variance is largely overlooked here. In the experiments, mini-batch based gradient is used.
> >
> > Regarding the overhead of the intermediate step, my earlier question—“Please measure and report this overhead quantitatively”—remains unanswered. Table 12 appears to present the overall communication cost, but it would be more informative to provide the explicit overhead of the intermediate step to clarify the trade-off involved.
> >
> > For the experimental evaluation, I still have concerns regarding generality; additional experiments on other domains are needed. I also agree with the other reviewers that more baselines with diverse sampling strategies should be included.

---

> > > ### Author Response · Authors · 2025-11-28
> > >
> > > We thank Reviewer hF4z for their assessment of our theoretical analysis and for helping us to strengthen the theoretical foundations of our work. Throughout our response, we refer to the revised version of the paper, which we have uploaded to OpenReview. For convenience, all major additions and revisions in the manuscript are highlighted in blue. We have expanded our analysis in the revision (highlighted in blue) to directly address concerns regarding Monte Carlo estimation, the stochastic mini-batch nature of FL as well as new baselines and domains included in our paper.
> > >
> > > >1. On Monte-Carlo estimation in the proof
> > >
> > > We acknowledge that proving the descent property (Theorem 4.4) strictly requires addressing the Monte Carlo approximation used in the update step. However, since the samples of $m$ devices across different rounds are IID, we can derive the expected descent of the step as shown in Eq. 5: $\mathbb{E}\delta f_{\tau+1/2}(m_{\tau+1})$. Thus, in Theorem 4.4 we provide guarantees that, on average, Algorithm 2 is able to successfully execute the procedure described in line 5. We kindly ask the reviewer to take note of the updated version of Theorem 4.4 in the revised version of the paper.
> > >
> > > We further clarify that the proof of Theorem 4.5 does not strictly require Monte Carlo estimation, as the update step explicitly models a single random sampling of $m$ clients without replacement. In Theorem 4.5, our goal is not merely to establish a descent property to verify the correctness of the algorithmic procedure, as was done in Theorem 4.4, but rather to provide full theoretical convergence guarantees for the algorithm. Consequently, we provide this guarantee with respect to the average gradient norm of the true objective function $f$, which is the standard form of convergence result for algorithms in the non-convex setting.
> > >
> > > >2. On obtaining the heterogeneity parameter $\rho$
> > >
> > > In practice, we do not need to know the exact value of $\rho$. If $\rho$ were known, we could indeed analytically determine the optimal $m$. However, the unavailability of $\rho$ is precisely why we designed the ISP procedure: it numerically approximates the optimal $m$ via Monte-Carlo estimation (Algorithm 2, Section 3.2) without requiring prior knowledge of data distribution. Regarding the theoretical threshold for $m$: while the theory mandates $m$ to be above a certain lower bound to guarantee descent, our experimental results align perfectly with this. As illustrated in Figure 2 and Figure 5, in the early stages of training, client gradients are largely co-aligned (effectively low local heterogeneity relative to the global gradient), which allows the algorithm to select a small $m$. As training progresses and the model approaches the optimum, the effective heterogeneity increases, prompting ISP to adaptively increase $m$. This dynamic behavior is fully consistent with the theoretical dependencies derived in our analysis.
> > >
> > >
> > > >3. On Mini-batch vs. Full-batch gradients
> > >
> > > The reviewer correctly notes that our theoretical analysis assumed a full-batch regime (Equation 1) to compare against distributed gradient descent. We treat the use of mini-batch stochastic gradients in experiments as a practical tool. In response to the feedback, we have extended our theoretical framework to the stochastic regime, adding Theorem F.5 (Stochastic Setting) and Theorem F.6 (Mini-batch Effect) in Appendix F.  Our new proofs explicitly model local updates with mini-batch size $b$ from a local dataset $B$. We derive a precise finite population correction factor $\frac{B-b}{b(B-1)}$ that governs the variance. We prove that the mini-batch noise introduces an additive term to the convergence bound $\mathcal{O}(\sigma \sqrt{\frac{B-b}{b(B-1)}})$. Crucially, this term is orthogonal to the communication savings from adaptive $m$. This rigorous derivation confirms that our method guarantees convergence even under partial mini-batch sampling, providing the missing theoretical justification for our experimental setup.
> > >
> > > Please, see our next comment where we address other questions regarding the experimental part.

---

> > > ### Author Response · Authors · 2025-11-28
> > >
> > > > 4. Overhead in the intermediate step
> > >
> > > We agree that dedicated discussion of communication overhead is necessary. Addressing the question, we add Section D.5, which analyzes communication overhead across all experiments considered. In this section we present a detailed quantitative breakdown, along with tables illustrating baseline overheads and relaxed parameter settings that reduce it.
> > >
> > > Our findings show that ISP’s intermediate-round overhead depends mainly on the parameters $\Delta$ and $m_{\tau+1/2}$​. Moderate adjusting these parameters has minimal impact on final model performance, while enabling substantial reduction of overhead. Table 15 summarizes our considerations. For example, using $\Delta = 100$ and $m_{\tau+1/2} = 80$ reduces the overhead to below 7% of total communication.
> > > | Method                                   | Communications | Overhead      | Test Loss | Accuracy |
> > > |-------------------------------------------|----------------|---------------|-----------|----------|
> > > | **ISP-Uniform**                           | 14,343         | × 1.430       | 0.464     | 0.836    |
> > > | ISP  m_{τ+1/2} = 80                        | 12,842         | × 1.311       | 0.461     | 0.840    |
> > > | ISP  m_{τ+1/2} = 60                        | 15,900         | × 1.252       | 0.467     | 0.845    |
> > > | ISP  Δ = 100                               | 12,652         | × 1.089       | 0.472     | 0.831    |
> > > | **ISP  Δ = 100, m_{τ+1/2} = 80**           | 12,924         | **× 1.069**   | 0.465     | 0.837    |
> > >
> > > In response to the Reviewer’s suggestions, we conduct several additional experiments that further broaden and strengthen our empirical evaluation.
> > >
> > > > 5. Other domains
> > >
> > > Addressing concerns on the need for additional experiments on other domains, we evaluate ISP under uniform client sampling in the text domain using the Shakespeare dataset from the LEAF benchmark [1]. The results and comprehensive discussion are included in Section D.1.1. Table 7 demonstrates the quantitative impact of dynamic selection of the number of participants. As shown, ISP consistently improves communication efficiency while maintaining the final model performance. Notably, this dataset features a large number of clients and highly heterogeneous user partitions; ISP remains effective under these challenging conditions. Full experimental details can be found in the corresponding section.
> > >
> > > | Method           | Communications (↓) | Test Loss (↓) | Accuracy (↑) | Perplexity (↓) |
> > > |------------------|--------------------|---------------|--------------|-----------------|
> > > | **Uniform**       | 18000             | 1.676         | **0.546**    | 5.35            |
> > > | **ISP-Uniform**   | **13015**         | **1.623**     | 0.541        | **5.07**        |
> > >
> > > > 6. Baselines
> > >
> > > We also add another advanced client selection method using Oort [2] strategy as an example. In contrast to other baselines considered in the main part, Oort explicitly addresses both statistical and system heterogeneity. Section D.1.2 discusses each of these contributions, the system heterogeneity in our codebase, and the experimental results. We briefly discuss the organization of system heterogeneity below.
> > >
> > > In our experimental setup, we selected a subset of clients, specifically a fraction of $p_1 = 0.1$, to introduce additional communication delays. We model these delays as the number of dummy training steps, compensating for the identical hardware resources. The number of additional steps for each selected client was sampled from a geometric distribution, denoted as $\xi \sim \mathrm{Geom}(p_2)$, where the success probability parameter $p_2$​ was set to 0.1 for our experiments.
> > >
> > > Table 8 demonstrates the results of ISP impact, showing reduced communication by ~20% while maintaining final accuracy and test loss. This demonstrates not only the addition of a new strong baseline, but also that ISP remains effective under system-level heterogeneity, where client training time varies across rounds.
> > > | Method         | Communications (↓) | Test Loss (↓) | Accuracy (↑) |
> > > |----------------|--------------------|---------------|--------------|
> > > | **Oort**       | 19500              | 0.478         | 0.834        |
> > > | **ISP-Oort**   | **15448**          | **0.414**     | 0.835        |
> > >
> > > ---
> > > We have addressed all raised concerns and hope that our clarifications fully resolve all questions. We are also open to provide any further details if needed.
> > >
> > > [1] Caldas, Sebastian, et al. "Leaf: A benchmark for federated settings."
> > >
> > > [2] Lai, Fan, et al. "Oort: Efficient federated learning via guided participant selection."

---

### Official Review · Reviewer_3BtW · 2025-10-31

**Soundness:** 3
**Presentation:** 3
**Contribution:** 3
**Rating:** 6
**Confidence:** 4

**Summary:**

This paper introduces Intelligent Selection of Participants(ISP), a dynamic mechanism that adaptively determines the optimal number of clients to participate in each communication round of Federated Learning (FL). ISP treats client number as a tunable variable and selects the minimal count needed to guarantee expected model improvement. This paper conduct experiments on CIFAR-10, Tiny-ImageNet, and a large real-world ECG dataset.

**Strengths:**

- This paper proposed the approach to determine the optimal number of participating clients per round. This adaptive viewpoint expands the optimization scope of FL and directly addresses communication bottlenecks.
- Extensive experiments on both standard benchmarks (CIFAR-10, Tiny-ImageNet) and a large-scale, real-world ECG dataset substantiate ISP’s practical relevance. The reported 30% communication reduction with comparable or better accuracy is a outcome.
- The authors include detailed ablations, for example, depth, windows, delta etc., theoretical derivations enhancing reproducibility.

**Weaknesses:**

- The proposed intermediate full-client communication step (Algorithm 2) contradicts the goal of communication efficiency, especially for large-scale networks. Though amortized over delta rounds, it still introduces a potential scalability concern.
- ISP partially mitigates communication costs, but its performance still depends on user-defined parameters, which may affect the results of dynamic selection.

**Questions:**

- For Table 2, please define the unit of measurement used in the *Communication* column. It’s unclear whether it represents the total number of communication rounds, total transmitted updates, or another metric.
- In Table 2, the average training time for ISP-FedCor appears about one hour longer than its baseline FedCor result. Could the authors clarify this discrepancy? Since ISP is expected to select fewer clients per round, a shorter overall training time would normally be expected. Please explain the underlying cause of the slower runtime.
- Regarding Figure 2, the plots suggest that ISP consistently outperforms its corresponding baselines. However, in Figure 2(b), we observe that as the number of clients increases, the model performance also improves across algorithms. Given that ISP often selects fewer clients, could the authors elaborate on how ISP maintains or enhances performance under reduced client participation?

---

> ### Author Response · Authors · 2025-11-21
>
> We thank Reviewer 3BtW for their time and assessment. Below, we address all the concerns. Throughout our response, we refer to the revised version of the paper, which we have uploaded to OpenReview. For convenience, all major additions and revisions in the manuscript are highlighted in blue.
>
> > W1: Intermediate full-client communication contradicts communication efficiency
>
> While Algorithm 2 presents the intermediate step as a full-client round for conceptual clarity, ISP does not obligatorily require full participation in practice. Table 12 demonstrates that the method remains stable - and even reduces total communication - when significantly fewer clients participate in the intermediate round. This directly models real large-scale FL environments where clients routinely fail to respond due to connectivity issues or resource constraints.
>
> Moreover, the intermediate step is invoked only once every $\Delta$ rounds, and $\Delta$ can be set large to further amortize its cost with no loss in stability (see Appendix E). Our ECG experiment with 2,000 clients (Table 4) (see also papers on AI ECG analysis [1,2]) reinforces this: despite large-scale participation, ISP delivers significant communication savings and only benefits from scalability.
>
> > W2: Performance depends on user-defined parameters
>
> Indeed, ISP introduces several hyperparameters; however, all of them are explicitly analyzed in Appendix E (Ablation Study). Importantly, each parameter has a structurally simple role: $\Delta$ controls invocation frequency and only affects overhead (E2); $\beta$ provides mild temporal smoothing with monotonic effects (E3); $N$ determines estimator variance and can be safely kept small (E5); $\omega$ governs search granularity and is robust except under extreme settings (E6). Each parameter is varied independently across wide ranges, and results consistently show stable, smooth behavior-performance does not exhibit sharp sensitivity or abrupt degradation.
> The default values used in our main experiments fall well within these stable regions, confirming the method's robustness.
>
> > Q1: Unit of measurement in Communication column
>
> In Definition 3.2, then in Section 5.1, and finally in Appendix C, we clarify that communication cost is measured by the number of client-to-server exchanges, as this reflects the primary bottleneck in federated learning due to transmitting large model payloads from distributed nodes. Specifically, both full and intermediate communication rounds are included in this calculation. We appreciate your comment and add a clear note near Table 2 to avoid any ambiguity regarding the communication metric.
>
> > Q2: Training time for ISP-FedCor vs FedCor
>
> We would like to clarify that there appears to be a misunderstanding. In Table 2, the average training time for FedCor is 5h 44m 05s, whereas for ISP-FedCor it is 4h 34m 57s. As expected, ISP-FedCor–by adaptively selecting fewer clients per round–achieves shorter overall training time compared to the baseline.
>
> However, as stated in the paper: “Only for the \textsc{POW-D} baseline, the \textsc{ISP} technique requires barely more communication but demonstrates a substantial increase in the target metrics that we attribute to the adversarial nature of client sampling in \textsc{POW-D}.”
> Indeed, POW-D client selection algorithm tends to prioritize clients with the highest validation loss, which, under high heterogeneity, may result in the selection of clients that degrade model performance -- close to Byzantine behavior in FL. To mitigate this issue, our proposed ISP method requires a larger number of participating clients, thereby improving the robustness and final performance of the trained model.
>
> It is also important to note that if POW-D were configured with a larger fixed number of clients per round, it could achieve final performance comparable to ISP - POW-D. However, this would come at the cost of substantially higher communication, since increasing the round-wise participant count directly increases communication overhead. We intentionally fixed the number of clients to m = 20 for all client-sampling baselines, following the careful analysis presented in Appendix, Section D.2 (Table 7), in order to provide a fair and consistent comparison.
> The POW-D case represents a trade-off where the modest communication increase is offset by meaningful performance gains, confirming ISP’s overall efficiency advantage.
>
> Please, see our next comment where we address other Questions.

---

> ### Author Response · Authors · 2025-11-21
>
> > Q3: How ISP maintains performance with fewer clients
> The reason ISP maintains or enhances performance while using fewer clients is that the method is explicitly designed to select the optimal number of participants sufficient for continued convergence at each stage of training.
>
> As shown in Figure 2, early in training, using a large number of clients is unnecessary because the model is far from the minimum and client gradients are largely co-aligned. As training progresses and the model approaches the optimum, client gradients become more diverse due to data heterogeneity, requiring more participants.
>
> We would like to note that this behavior is fully aligned with the theory established. Our analysis uncovers the physical intuition driving the method: the required participation $m$ scales with data heterogeneity ($\rho$) rather than the total network size ($M$). In simple terms, we provide a formal justification that in regimes with moderate heterogeneity (low $\rho$), the theoretical need for massive participation disappears, simplifying to $m = \mathcal{O}(\rho)$. This insight fundamentally shifts the selection of $m$ from a heuristic guess to a principled decision governed by the complexity of the loss landscape.
>
> —
>
> Overall, we would like to emphasize that the identified weaknesses W1 and W2 have been clarified: as demonstrated, they do not negatively impact convergence or runtime. Questions Q1-Q3 have been addressed with explicit references to existing content and minor clarifications added where needed. Additionally, in response to the feedback, we have substantially strengthened the contribution by adding formal convergence guarantees (Appendix F), which not only establish rigorous theoretical foundations but also explain the fundamental trade-offs underlying ISP’s empirical success. We believe these additions, combined with the clarifications provided, have significantly enhanced the paper’s contribution.
>
> [1] Jing E. et al. ECG heartbeat classification based on an improved ResNet‐18 model
>
> [2] Han C., Shi L. ML-ResNet: A novel network to detect and locate myocardial infarction using 12 leads ECG

---

> > ### Comment · Reviewer_3BtW · 2025-11-22
> >
> > Thank you for provide the grateful feedback. But I will keep the score for this time.

---

### Official Review · Reviewer_42v7 · 2025-11-01

**Soundness:** 2
**Presentation:** 2
**Contribution:** 2
**Rating:** 2
**Confidence:** 3

**Summary:**

This paper proposes ISP (Intelligent Selection of Participants), an adaptive mechanism for federated learning (FL). ISP dynamically determines the optimal number of clients per training round to enhance communication efficiency without compromising model accuracy. Unlike existing FL methods, which assume a fixed number of participants, ISP formulates the round-wise client count selection as a constrained optimization problem, selecting the minimum number of clients needed to achieve the expected loss reduction. The authors validate ISP across diverse setups, including vision transformers, real-world ECG classification, and gradient-compressed training. In these cases, ISP consistently achieves communication savings of up to 30% without degrading performance.

However, the work is limited by insufficient theoretical support, unaddressed hyperparameter sensitivity, and incomplete comparisons with the latest baselines. The computational overhead from intermediate communications also requires further optimization for resource-constrained environments.

**Strengths:**

1. ISP formalizes the round-wise choice of the number of participants as a constrained problem, selecting the smallest value that achieves expected loss decrease.
2. The framework is compatible with popular FL algorithms and requires no changes to client optimizers, enabling easy integration with standard FL pipelines.

**Weaknesses:**

--Unmitigated ISP overhead restricts edge use. The Monte-Carlo approach in ISP introduces heavy computational overhead, which is not fully mitigated and may limit applicability in resource-constrained edge environments.

--No comprehensive analysis of ISP hyperparameter impact. ISP relies on multiple hyperparameters (e.g., window Δ, momentum β, resolution ω), but the paper lacks a comprehensive analysis of how these parameters affect performance across different FL scenarios.

--Limited Theoretical Justification. The optimization formulation for client count selection is primarily validated empirically, yet theoretical analysis on convergence guarantees and the trade-off between communication efficiency and model accuracy is insufficient.

--A lack of comparison with advanced baselines. The work compares ISP with classic and some state-of-the-art client sampling methods but overlooks recent advanced adaptive FL frameworks, making it hard to assess ISP’s competitiveness against the latest techniques.

**Questions:**

Please see weaknesses.

---

> ### Author Response · Authors · 2025-11-21
>
> We thank Reviewer 42v7 for their feedback. We answer all the concerns below. In our response, we refer to the revised version of the paper, uploaded to OpenReview. All major additions and revisions in the manuscript are highlighted in blue.
>
> > W1: Monte Carlo overhead
>
> The Monte Carlo overhead in ISP is highly controllable and minimal in practice. The computational cost is determined by the hyperparameter $N$, which can be adjusted based on available resources. Thus, Appendix E.5 (Table 16) demonstrates that even under strong data heterogeneity, very small values of $N$ preserve most of ISP’s communication savings while causing only minor performance degradation.
>
> Critically, ISP also supports server-side estimation using auxiliary data when available (Section 3, Problem 3). This shifts the Monte Carlo evaluation entirely to the server–typically a more powerful machine-eliminating any computational burden on resource-constrained edge devices. In this mode, clients only perform standard local training and communication, with no additional overhead compared to baseline federated learning.
>
> We also note that Table 2 presents a runtime comparison demonstrating that despite the intermediate estimation step, ISP substantially reduces total training time compared to baselines. In large-scale settings with many clients (Table 4), these time savings increase even further, confirming that it does not impede practical deployment.
>
> >W2: No comprehensive hyperparameter analysis
>
> We respectfully disagree. Appendix E provides a detailed ablation study for all ISP hyperparameters: $\Delta$ (update interval), $\beta$ (momentum), $\omega$ (resolution), and $N$ (Monte Carlo depth).
> Importantly, each parameter has a structurally simple role: $\Delta$ controls invocation frequency and only affects overhead; $\beta$ provides mild temporal smoothing with monotonic effects; $\omega$ governs search granularity and is robust except under extreme settings; $N$ determines estimator variance and can be safely kept small (Appendix E.5, Table 16).
>
> Each parameter is varied independently across wide ranges, and results consistently show stable, smooth behavior-performance does not exhibit sharp sensitivity or abrupt degradation.
> The default values used in our main experiments fall well within these stable regions, confirming the method's robustness.
>
> >W3: Theoretical Justification
>
> We sincerely appreciate the reviewer’s emphasis on theoretical rigor. We dedicated the revision period to formalizing the mathematical foundation of ISP.
>
> To this end, we have introduced Theorem 4.4, which derives an explicit lower bound for the participation size $m_{\tau+1}$. This bound is the precise condition required to guarantee a strict reduction in the global loss function at each step. Furthermore, Theorem 4.5 proves that ISP maintains the standard non-convex convergence rate of $\mathcal{O}(1/T)$. This confirms that our dynamic reduction of clients does not come at the cost of stability or asymptotic convergence speed compared to fixed-size baselines.
>
> Perhaps most importantly, our analysis uncovers the physical intuition driving the method: the required participation $m$ scales with data heterogeneity ($\rho$) rather than the total network size ($M$). In simple terms, we provide a formal justification that in regimes with moderate heterogeneity (low $\rho$), the theoretical need for massive participation disappears, simplifying to $m = \mathcal{O}(\rho)$. This insight fundamentally shifts the selection of $m$ from a heuristic guess to a principled decision governed by the complexity of the loss landscape.
>
> > W4: Comparison with advanced baselines
>
> ISP addresses the problem of dynamically determining how many clients participate per round. It is largely unexplored in the literature. Existing "advanced adaptive FL" methods focus on which clients to select (e.g., importance sampling, utility-based selection, diversity-based sampling) or on optimizer-level adaptations, but they all assume a fixed participant count. These approaches are complementary to ISP rather than directly comparable. ISP can be combined with any client selection strategy, as demonstrated in Table 2 where ISP improves performance across Uniform, POW-D, FedCor, FedCBS, and DELTA sampling methods.
>
> Appendix D.4 (Table 11) includes a comparison with the closest available baseline: AdaFL-inspired linear client-count schedule. Results show that predetermined schedules are suboptimal under heterogeneity, while ISP consistently improves both communication and accuracy.
>
> ---
> We believe these additions combined with the clarifications have significantly strengthened the contribution. ISP addresses a novel and practically important dimension in federated learning, is supported by both theory and extensive experiments, and operates orthogonally to existing methods without requiring client-side changes. We respectfully ask the reviewer to reconsider the evaluation in light of this rebuttal.

---

### Official Review · Reviewer_NQb9 · 2025-11-03

**Soundness:** 2
**Presentation:** 3
**Contribution:** 2
**Rating:** 6
**Confidence:** 4

**Summary:**

The paper tackles an under-explored question in federated learning which is finding out how many clients should participate per round. Existing methods fix this number, focusing only on which clients to sample. The authors propose Intelligent Selection of Participants (ISP), an adaptive mechanism that dynamically adjusts client count based on observed training progress to improve communication efficiency. ISP periodically runs a lightweight intermediate round to estimate how model performance changes with different client counts. It then selects the smallest number of clients that still improves the loss and smooths updates over time to prevent oscillation. The method integrates with standard FL algorithms (FedAvg, FedProx, SCAFFOLD) and techniques like client sampling and gradient compression. Experiments on CIFAR-10, Tiny-ImageNet, and real ECG data show fewer communication rounds without accuracy loss.

**Strengths:**

- **S1:** Clearly motivated and easy to integrate into existing FL systems.
- **S2:** Consistent gains in communication efficiency across tasks and datasets.
- **S3:** Works with client selection and compression methods.
- **S4:** Thorough experimental coverage, including a large ECG setup.

**Weaknesses:**

- **W1:** Requires synchronized or full-client intermediate rounds, which limit scalability.
- **W2:** No theoretical analysis of convergence or stability.
- **W3:** ISP highlights a neglected dimension of FL optimization: dynamically adjusting client count to balance efficiency and performance. While conceptually straightforward, it’s practical and general, requiring no client-side changes. The contribution is incremental but relevant to real-world FL deployments.
- **W4:** Modest conceptual novelty given prior adaptive participation work.

**Questions:**

- **Q1:** How does ISP behave under highly variable client availability, such as mobile or cross-device FL?
- **Q2:** How sensitive is the algorithm to its tuning parameters such as particularly the interval between updates and the smoothing coefficient?
- **Q3:** Could the intermediate probing phase be replaced by a lighter-weight estimation, such as tracking gradients or validation loss trends?
- **Q4:** How does the approach compare to existing adaptive participation heuristics, such as linear decay or reinforcement-learning-based scheduling?
- **Q5:** Could the approach handle asynchronous updates, where clients finish at different times?

---

> ### Author Response · Authors · 2025-11-21
>
> We thank Reviewer NQb9 for their detailed response. Below we address all concerns. Throughout our response, we refer to the revised version of the paper, which we have uploaded to OpenReview. For convenience, all major additions and revisions in the manuscript are highlighted in blue.
>
> > W1: Full-client intermediate rounds
>
> Importantly, ISP works even without fully synchronized or full-client participation in the intermediate round. As demonstrated in Table 12, the method remains stable even when significantly fewer clients contribute updates, and its performance does not degrade. In fact, we even observe improved overall communication efficiency - and in some cases better final performance when only 80 out of 100 clients participate in the $ \tau + 1/2$ step. This indicates that ISP is naturally robust to partial participation and does not rely on strict synchronization, which supports its scalability in practical large-scale or unstable FL environments.
>
> Furthermore, our ECG experiment contains 2,000 clients (see Table 4), which demonstrates that ISP scales to large federations and provides even larger communication savings. These results together indicate that ISP does not depend on perfectly synchronized full-client rounds; instead, it naturally tolerates partial responses and remains effective at scale.
>
> > W2: Theoretical guarantees
>
> We fully agree that theoretical grounding is essential. Given that our extensive ablation studies (Appendix C & D) already confirm the method’s empirical robustness, we prioritized the rebuttal period to rigorously formalize ISP. We have added Theorem 4.4, which derives the explicit lower bound for participation $m_{\tau+1}$ sufficient to guarantee a strict reduction in the global loss function. Building on this, Theorem 4.5 establishes that ISP preserves the standard non-convex convergence rate of $\mathcal{O}(1/T)$, mathematically proving that dynamic client reduction does not compromise the learning trajectory or asymptotic stability compared to fixed-size baselines.
>
> Crucially, we reveal the physical meaning behind these bounds. Thus, the required participation $m$ scales with data heterogeneity ($\rho$) rather than the total network size ($M$). Specifically, we provide a formal justification for our method’s behavior: in regimes with moderate heterogeneity (low $\rho$), the theoretical necessity for massive participation vanishes and it simplifies to $m = \mathcal{O}(\rho)$. This insight moves the choice of $m$ from heuristic guessing to a principled function of the loss landscape’s complexity.
>
> > W3 & W4: Conceptual novelty
>
> We thank Reviewer for pointing out that our contribution is relevant. However, we do not agree that novelty is modest. While extensive prior literature focuses on qualitative sampling (i.e., “which” clients to select), ISP pioneers the optimization of participation magnitude (i.e., “how many”), effectively transforming the batch size $m$ from a static hyperparameter into a dynamic control variable derived explicitly from our convergence analysis (see answer to W2). This represents a fundamental conceptual shift — moving from heuristic-based to theoretically-grounded participation — rather than a minor adjustment. Crucially, this contribution is orthogonal to existing strategies; as shown in Table 2, ISP acts as a universal accelerator that significantly enhances SOTA baselines (e.g., POW-D, FedCor, FedCBS, and DELTA) without altering their core logic. Thereby, ISP unlocks a distinct and critical dimension of efficiency in FL.
>
> Please, see our next comment where we address Questions.

---

> ### Author Response · Authors · 2025-11-21
>
> > Q1: Variable client availability
>
> Table 12 directly addresses this concern: it demonstrates ISP’s robustness when clients fail to respond in the intermediate round. Even with up to 40% missing updates, ISP maintains both communication savings and final performance. This naturally models mobile/cross-device settings where clients drop out.
>
> For environments with severe instability, ISP supports an optional server-side resampling strategy using Monte Carlo estimation on available server data (Section 3, Problem 3). This requires no additional communication and substantially mitigates the impact of unpredictable client availability patterns.
>
> >Q2: Sensitivity to tuning parameters
>
> Appendix E provides a detailed ablation study on all key hyperparameters: the update interval $\Delta$, momentum coefficient $\beta$, Monte Carlo depth $N$, and resolution $\omega$. Results show that ISP is not particularly sensitive to any single parameter-across broad value ranges, the method consistently maintains both communication efficiency and final accuracy.
>
> >Q3: Lighter-weight estimation
>
> We explored this direction by implementing a gradient- and loss-based heuristics that tracked loss history to adjust client count dynamically. However, this approach proved highly sensitive to hyperparameters and failed to achieve meaningful communication savings, often defaulting to large participant counts. Under strong data heterogeneity, validation loss and gradients alone are insufficiently informative for reliable decision-making. Our experiments indicate that specialized estimation mechanisms–such as ISP's intermediate probing–are necessary to handle heterogeneous FL scenarios effectively.
>
> >Q4: Comparison to adaptive participation heuristics
>
> Appendix, Section D.4 (Table 11) presents a comparison with linear scheduling heuristics for increasing client count. Results demonstrate that linear strategies are less efficient than ISP, highlighting the need for an intelligent, criterion-based approach rather than predetermined schedules.
>
> Regarding RL-based methods, existing work primarily addresses which clients to select rather than how many [1,2,3,4]. Moreover, RL approaches introduce substantial computational overhead–requiring additional policy updates - and face well-known challenges such as training instability and convergence difficulties. For the problem we address, RL is inherently mismatched: our task requires a stable, low-overhead decision criterion that can be efficiently evaluated in real-time, whereas RL incurs significant training and inference costs without clear benefit. That said, we remain open to discussing specific RL implementations if the reviewer believes there are directly relevant prior works we may have overlooked.
>
> >Q5: Asynchronous updates
>
> ISP is designed for synchronous FL and does not address asynchronous aggregation. Our contribution focuses on a distinct dimension: dynamically selecting how many clients to involve. Our work preserves the partial-participation setting adopted in prior works ([5], [6]). We view asynchronous analysis as a fundamentally different problem. Moreover, based on known lower bounds on convergence ([7], Table 1), in an asynchronous setup with heterogeneous data distributions it is effectively necessary to wait for the slowest transmission. This implies that if our algorithm were to wait for the last client at this fixed final communication moment, it would not affect the potential convergence guarantees in such a setup.
>
> That said, ISP's core mechanism could be adapted to asynchronous settings. We can make the relaxation of an asynchronous regime by a setup in which there exists a time point at which full communication takes place. However, we highlight that this extension is outside the current scope.
>
> ---
> Overall, we addressed all raised concerns. We clarified that the requested experiments were largely established (Tables 2, 4, 11, 12; Appendix E). Most critically, we strengthened the paper’s foundation by deriving rigorous convergence guarantees (Section 4, Appendix F) in response to W2. This bridges the gap between our empirical performance and theory, elevating the work to a fully grounded optimization framework. We are confident that these additions significantly reinforce the paper’s impact and validity. We respectfully ask the reviewer to reconsider the evaluation in light of these clarifications.
>
> [1] Rjoub, Gaith, et al. Trust-augmented deep reinforcement learning… Information Systems Frontiers 2024
>
> [2] Zhang, Sai Qian,et al. A multi-agent reinforcement learning approach... AAAI 2022
>
> [3] Chen, Weikang, et al. Dynamic fair federated learning… DOCS 2023
>
> [4] Pan Z. et al. RFCSC: Communication efficient reinforcement federated learning… Neurocomputing, 2025
>
> [5] Wang et al. Tackling the objective... NeurIPS 2020
>
> [6] Cho et al. Towards understanding biased client... AISTATS 2022
>
> [7] Tyurin, Richtarik. Optimal Time Complexities of Parallel Stochastic... NeurIPS 2023

---

> > ### Comment · Reviewer_NQb9 · 2025-11-27
> >
> > Thanks for the detailed responses! The revised paper is satisfactory, and I will keep my positive rating.

---

### Comment · Area_Chair_sCTy · 2025-11-27

Dear Reviewers,

Thank you for the time and effort you have dedicated to reviewing this paper and providing thoughtful feedback. The authors have now submitted their responses to your comments. I kindly ask that you engage in the discussion with them and assess whether your concerns and questions have been fully addressed before the December 2 deadline.

Please also keep in mind that the author–reviewer relationship is reciprocal; the engagement you offer here reflects the same level of consideration you would expect when you are on the author side.

Thank you for your continued support and cooperation.

Best regards,
AC

---

### Author Response · Authors · 2025-12-03

Dear Area Chair,

We know this review cycle has been unexpectedly difficult, and we appreciate all the efforts to keep things moving forward. Given that, we want to share a brief summary of where our paper stands after the rebuttal to help with final assessment.

During the rebuttal, we focused on strengthening the paper based on the reviewers' feedback. We introduced a theoretical analysis to support our ISP mechanism in Section 4 and Appendix F. Additionally, we conducted new experiments, including evaluations on a new domain and comparisons against additional baselines, to further validate the robustness and communication efficiency of our method (Appendix D.1, D.5).

Regarding the review process, we received constructive engagement from the reviewers.
Reviewers NQb9 and 3BtW maintained a positive score and expressed satisfaction with our revisions.
Reviewer hF4z engaged in the most detailed discussion. We provided extensive responses to their questions and suggestions, including the requested additional results. We believe we addressed their concerns, however, we were unable to receive their final confirmation due to the sudden system restrictions.
Unfortunately, due to the circumstances, Reviewer 42v7 did not participate in the discussion phase.

We believe that with the added theoretical grounding and extended experimental validation, our work effectively addresses the key challenges in federated learning communication efficiency.

Best regards,

The Authors

---

### Meta-Review · Area_Chair_UK8f · 2026-01-06

**Summary:**

This paper addresses the under-explored problem of dynamically selecting the number of clients participating in each federated optimization round, rather than the well-studied problem of which clients to select. The method periodically runs intermediate communication rounds to estimate expected loss reduction for different participation levels, then selects the minimum number of clients ensuring model improvement. The paper provides theoretical convergence analysis and demonstrates up to 30% communication savings across different datasets and FL methods.

The authors provided a substantial revision during the rebuttal, adding:
- Theorem 4.4, 4.5 with complete proof establishing O(1/T) convergence rate and conditions on minimum participation.
- new experiments on text datasets
- new baselines (OORT)
- several clarifications regarding hyper-parameters, and communication saving.

The revision substantially improved the paper. While the remaining theoretical gap of the MC estimation analysis is a legitimate concern, it does not fundamentally undermine the contribution of the paper. The empirical evidence shows that on small- or moderate-scale datasets (CIFAR10, Tiny-ImageNet, 2K-client ECG); larger-scale and more realistic experiments (e.g., inaturalist with 10K clients) could have strengthened the paper and mitigated concerns about the scalability of the method.

Personally, the main concern of the method is its applicability in real-world scenarios because of the intermediate full-participation rounds, which require communicating with all clients (unrealistic). Any relaxation (communicating with a portion of the clients) would be affected by the heterogeneity, causing a biased estimate. This effect is evaluated in Table 16, varying the number of participating clients from 100 to 20 (no details on the dataset and setting used are provided, but I assume this is cifar 10), showing that at least 60% of the clients are necessary. Furthermore, there is no formal guarantee that partial intermediate participation preserves convergence properties.

A related concern is the noise of the evaluation. The proposed method should reach at least the same performance as the baseline, which is not always the case from the tables and plots. A better communication metric would be the number of communication bits necessary to reach the 80%, 90% and 100% of the baseline +- std variation. Figure 2b shows unexpected instability in the Uniform baseline (m=400) around rounds 160-200, which warrants clarification from the authors regarding whether this reflects inherent variance, hyperparameter issues, or dataset characteristics.

Rev 42v7 also asks for adaptive baselines; it is unclear what specifically they refer to. The authors interpret it as client selection methods and added ORT. However, this could also reasonably refer to optimization methods designed for heterogeneity (SCAFFOLD, FedDyn, FedSAM, FedProx, Momentum). While the paper claims ISP is compatible with SCAFFOLD/FedProx, this is not empirically demonstrated. Experiments showing ISP's effectiveness when combined with these optimizers would strengthen the contribution, as these are commonly use such methods in heterogeneous settings.

Finally, I believe that the paper framing around communication efficiency may overstate the practical benefit of the contribution, while a more accurate characterization should be "automatic tuning of client participation," as ISP removes the need to cross-validate the participation hyperparameter at the cost of full-participation intermediate rounds.

**Reviewer Concerns:**

see above

**Reviewer Scores:**

- NQb9: 6
- 42v7: 2 -> 4
- 3BtW: 6
- hF4z: 4 -> 6

---

### Decision · Program_Chairs · 2026-01-26

Reject